

# Marine boundary layer aerosol in Eastern North Atlantic: seasonal variations and key controlling processes

Guangjie Zheng[1], Yang Wang[1], Allison C. Aiken[2], Francesca Gallo[2], Mike Jensen[1], Pavlos Kollias[1,3], Chongai Kuang[1], Edward Luke[1], Stephen Springston[1], Janek Uin[1], Robert Wood[4], and Jian Wang[1,*]

[1] Environmental and Climate Science Department, Brookhaven National Laboratory, Upton, New York, USA
[2] Earth System Observations, Los Alamos National Laboratory, Los Alamos, New Mexico, USA
[3] School of Marine and Atmospheric Sciences, Stony Brook University, State University of New York, Stony Brook, New York, USA
[4] Department of Atmospheric Science, University of Washington, Seattle, USA

*Correspondence to*: J. Wang (jian@bnl.gov)

**Abstract**

The response of marine low cloud systems to changes in aerosol concentration represents one of the largest uncertainties in climate simulations. Major contributions to this uncertainty derive from poor understanding of aerosol under natural conditions

and the perturbation by anthropogenic emissions. The Eastern North Atlantic (ENA) is a region of persistent but diverse marine boundary layer (MBL) clouds, whose albedo and precipitation are highly susceptible to perturbations in aerosol properties. In this study, we examine MBL aerosol properties, trace gas mixing ratios, and meteorological parameters measured at the Atmospheric Radiation Measurement Climate Research Facility's ENA site on Graciosa Island, Azores, Portugal from 2015 to 2017. Measurements impacted by local pollutions on Graciosa Island and during occasional intense biomass burning and dust

events are excluded from this analysis. Submicron aerosol size distribution typically consists of three modes: Aitken (At), Accumulation (Ac), and Larger Accumulation (LA) modes, with average number concentrations (denoted as $N_{At}$, $N_{Ac}$ and $N_{LA}$ below) of 330, 114, and 14 cm$^{-3}$, respectively. $N_{At}$, $N_{Ac}$ and $N_{LA}$ show contrasting seasonal variations, suggesting different sources and removal processes. $N_{LA}$ is dominated by sea spray aerosol (SSA), and is higher in winter and lower in summer. This is due to the seasonal variations of SSA production, coalescence scavenging, and dilution by entrained free troposphere (FT) air.

In comparison, SSA typically contributes a relatively minor fraction to $N_{At}$ (10 %) and $N_{Ac}$ (21 %) on an annual basis. In addition to SSA, sources of Ac mode particles include entrainment of FT aerosols and condensation growth of At mode particles inside MBL, while coalescence scavenging is the major sink of $N_{Ac}$. The observed seasonal variation of $N_{Ac}$, being higher in summer and lower in winter, generally agrees with the estimate based on the major sources and sink. $N_{At}$ is mainly controlled by entrainment of FT aerosol, coagulation loss, and growth of At mode particles into Ac mode size range. Our calculation suggests

besides the direct contribution from entrained FT Ac mode particles, growth of entrained FT At mode particles in the MBL also represent a substantial source of cloud condensation nuclei (*CCN*), with the highest contribution potentially reaching nearly 60 % during summer. The growth of At mode particles to *CCN* size is expected a result of the condensation of sulfuric acid from dimethyl sulfide oxidation, suggesting that ocean ecosystems may have a substantial influence on MBL *CCN* populations in ENA.





## 1 Introduction

Low clouds, especially stratocumulus, are the dominant cloud type in terms of spatial coverage of the Earth's surface, and are of vital importance to the Earth's climate (Wood, 2012). Major climate effects of low clouds derive from their reflection of solar radiation (Seinfeld and Pandis, 2016). The key parameters for quantifying climate effects of low clouds are the albedo (i.e., cloud
reflectivity) and the cloud coverage, both of which are particularly sensitive to perturbations of aerosols. The concentration of cloud condensation nuclei (*CCN*) strongly influences the number concentration and sizes of cloud droplets and therefore the effective albedo of low clouds (i.e., first indirect effect of aerosol) (Twomey, 1974; Seinfeld and Pandis, 2016; Dong et al., 2015), especially in clean environments such as remote marine boundary layer (MBL) (Reutter et al., 2009). In addition, *CCN* concentration and aerosol size distribution also influence cloud amount by impacting drizzle formation and precipitation (i.e.,
second indirect effect of aerosol) (Albrecht, 1989; Liu and Daum, 2004; Liu et al., 2006; Wood, 2005; Rémillard et al., 2012; Dong et al., 2014).

Currently, the aerosol indirect effects of marine low cloud systems remain one of the major uncertainties in climate change simulations (Lohmann and Feichter, 2005; Bony and Dufresne, 2005; Bony et al., 2006; Wood, 2012). This large uncertainty is
to a large degree a result of the incomplete understanding and therefore representations of aerosol properties, and the response of marine low clouds to aerosol changes. Therefore, it is imperative to understand MBL aerosol properties under natural conditions, the perturbation due to anthropogenic emissions, and the underlying controlling processes. The properties of aerosols in remote MBL can be influenced by a variety of processes, including entrainment from the free troposphere (FT), production of sea spray aerosol (SSA), processing of aerosol particles both inside clouds and in clear air, depositions, and horizontal advection (Quinn
and Bates, 2011; Wood et al., 2012). Previous studies (O'Dowd et al., 2004; Clarke et al., 2013; Quinn et al., 2017; Wood et al., 2017; Pierce et al., 2015; Prather et al., 2013; Russell et al., 2010; Sanchez et al., 2018) have greatly advanced our understanding of MBL aerosols, especially in the relative contributions of SSA versus long-range transported pollution in terms of *CCN* budget (Blot et al., 2013; Clarke and Kapustin, 2010; Clarke et al., 2013; Quinn et al., 2017), and the removal of *CCN* by coalescence scavenging (Wood et al., 2012). However, we are still lacking a quantitative understanding of the controlling processes sufficient
to serve as a reliable foundation for developing global climate model parameterizations and representations that will adequately simulate aerosol in past, current, and future climates. The relative importance, the influence on different particle size ranges, and spatiotemporal variations of these processes are still not well quantified.

The Eastern North Atlantic (ENA) is a region of persistent but diverse subtropical MBL clouds (Wood et al., 2015). Aerosols
arriving in ENA are of diverse origins, varying from marine clean air masses to air masses that are strongly influenced by continental emissions from North America or northern Europe (O'Dowd and Smith, 1993; Wood et al., 2015). As a result, ENA is among the regions with strong but uncertain aerosol indirect forcing (Carslaw et al., 2013). The atmosphere in ENA has been studied in several field campaigns, including the North Atlantic Regional Experiment (NARE) campaign during 1991 to 2001 (http://www.igacproject.org/activities/north-atlantic-regional-experiment-nare-1991-2001), the Atlantic Stratocumulus Transition
Experiment (ASTEX) during June 1992, the 2nd Aerosol Characterization Experiment (ACE-2) during summer 1997, and the Clouds, Aerosol, and Precipitation in the Marine Boundary Layer (CAP-MBL) campaign (www.arm.gov/sites/amf/grw) from May 2009 to December 2010. However, they are either more focused on other subjects (e.g., ozone chemistry for NARE (Parrish et al., 1998) and cloud properties for CAP-MBL (Rémillard et al., 2012; Wood et al., 2015; Rémillard and Tselioudis, 2015)), or are short-term campaigns (e.g., ACE-2 (Raes et al., 2000), ASTEX (Albrecht et al., 1995)). To our knowledge, the variation of
aerosol properties and their controlling processes have not been systematically studied using long term observation.



Recently, a permanent ENA site was established by the Department of Energy Atmospheric Radiation Measurement (ARM) Climate Research Facility on Graciosa Island in the Azores, Portugal, providing an invaluable opportunity to study MBL aerosol properties and their interactions with low clouds. In this study, we examine the long-term pattern of aerosol properties, trace gas

mixing ratios, and meteorological parameters measured at the ARM ENA site from 2015 to 2017 (section 2). The characteristics of the aerosol properties and their seasonal variations are summarized (section 3). The governing equations of number concentration are established for different modes of MBL aerosol at the ENA site (section 4). Subsequently, the seasonal variations of aerosol properties for different particle size modes are explained using key processes identified (section 5 and 6). Finally, we present an overall picture of the processes that drive MBL aerosol properties in ENA, and the implications are

discussed (section 7).

## 2 Measurement

### 2.1 Measurement overview

Measurements of trace gases, meteorological parameters, aerosol and cloud properties are conducted at the ENA site, located on

Graciosa Island in the Azores, Portugal (39° 5' 30" N, 28° 1' 32" W, 30.48 m above mean sea level). The ENA observation site was initially set up in late 2013, with additional measurements added subsequently. The primary measurements used in this study and the available time periods are listed in Table 1. The measurements of trace gases (e.g., CO) and aerosol properties were first filtered for local sources (see Supporting Information (SI) section S1). All measurements are then averaged into 1-hour intervals. Here we use three-years of data from Jan. 2015 to Dec. 2017 to show the long-term variations and correlations among different

parameters. For evaluation of the contributions of different controlling processes (section 4), one-year data from Sept. 2016 to Aug. 2017 are used, during which period most of the measurements are available.

### 2.2 Data corrections and derivations

### 2.2.1 Optical properties

Aerosol absorbing ($B_{abs}$) and scattering ($B_{sca}$) coefficients are measured by a three-wavelength PSAP and a Nephelometer, respectively (Table 1). These two instruments share a common inlet, and the 50 % cut size of the inlet switches between 1 and 10 μm every hour (Springston, 2016). The corresponding $B_{sca}$ and $B_{abs}$ are denoted by "PM$_1$" and "PM$_{10}$", respectively. In addition, properties of coarse mode ($1 < D_p < 10$ μm) aerosols, PM$_c$, were derived by the difference between PM$_{10}$ and PM$_1$. For example, "PM$_c$ $B_{sca}$" refers to the difference between PM$_{10}$ $B_{sca}$ and PM$_1$ $B_{sca}$ hereafter, and PM$_c$ $B_{abs}$ is defined similarly.

The mass flow calibration and filter loading correction are already applied to the PASP data in the ARM data archive (Springston, 2016). In this study, additional corrections of contribution due to scattering for $B_{abs}$ (Bond et al., 1999; Virkkula et al., 2005; Virkkula, 2010; Costabile et al., 2013), and truncation and angular illumination for $B_{sca}$ (Anderson and Ogren, 1998; Müller et al., 2011) are applied, and the procedure is detailed in SI section S2. The corrected PM$_1$ $B_{sca}$ shows strong correlation (correlation

coefficient being 0.84) with the volume of PM$_1$ derived from UHSAS size distribution (Fig. S2).



### 2.2.2 Cloud and MBL properties

The MBL height, $H_{MBL}$, is derived from the backscatter signal from the Ceilometer CL31 (Morris, 2012). Briefly, it is determined from the gradient of an idealized backscatter profile, the parameters of which from fitting of the observed profile (Eresmaa et al., 2006). As the first boundary layer height given in the ceilometer data product is usually the surface layer (Lewis
and Schwartz, 2004) below 100 m (Münkel et al., 2007; Emeis et al., 2007; Emeis et al., 2008; Haeffelin et al., 2012; Morris, 2012), $H_{MBL}$ is chosen as the highest boundary layer height below 3 km (Zhou et al., 2015; Rémillard and Tselioudis, 2015; Rémillard et al., 2012).

Cloud thickness $h$ is derived by combining $H_{MBL}$ and the cloud base height derived from CL31 data. In ENA, $H_{MBL}$ usually
represents the top height of boundary layer clouds (Rémillard et al., 2012). When multiple layers of clouds are detected, the layers with cloud-base heights higher than $H_{MBL}$ are first excluded, after which the highest layer is chosen to exclude potential influence of near-ground thin clouds. The cloud thickness $h$ is then defined as the difference between $H_{MBL}$ (cloud top) and the base height of the chosen cloud. The value of $h$ derived using the above approach is in general agreement with previous observations (Rémillard et al., 2012).

The cloud fraction, $p_{cloud}$, is determined by the detection status information from the ceilometer (Morris, 2012). It is equal to the fraction of time with a detected boundary layer cloud base, or a determined full obscuration. Precipitation rate at cloud base is retrieved from the vertically pointing K-band cloud radar (Atmospheric Radiation Measurement Climate Research Facility, 1990) and the Ceilometer CL31 (Morris, 2012) following the method of O'Connor et al. (2005).

### 3. Seasonal variation in synoptic conditions, trace gas mixing ratios, and aerosol properties in ENA

### 3.1 Air mass origin

One major source of MBL aerosol in ENA is the entrainment of the FT air, which can contain particles from long-range transport of continental pollutions, and those formed through new particle formation (NPF) in FT (Quinn and Bates, 2011; Sanchez et al., 2018). To examine the contribution due to continental emissions and its seasonal variation, we analyze the back trajectories of air
masses arriving at the ENA site. The cluster analysis results of 4 representative months from Sept. 2016 to Aug. 2017 (i.e., the main study period, see section 2.1) are shown in Fig. 1. Results from other periods from 2015 to 2017 are similar (not shown).

Most of the Air masses arriving at the ENA site can be classified as one of the four clusters originated from North America, northern Europe, the Arctic, and the recirculating flow around the Azores high, respectively (O'Dowd and Smith, 1993; Wood et
al., 2015). Among these clusters, the Azores high air masses usually linger within the MBL, as indicated by their stable and low-level trajectories (e.g., blue and red trajectory clusters in Fig. 1b). In comparison, other air masses usually undergo long-range transport within the FT before descending into MBL. In addition, some air masses originating in the continental boundary layer were lofted up, and then subsided into MBL within 10 days (e.g., blue trajectory cluster in Fig. 1c).

The percentage of occurrence for each cluster shows strong seasonal variations (Fig. 1). During fall (Fig. 1c) and winter (Fig. 1d), air masses influenced by anthropogenic emissions from North American air (red lines) dominate, with the influence of clean maritime flow and northern European flow. In spring (Fig. 1a), contribution from Arctic/northern Europe air masses is more pronounced than during other seasons. For the summer months (Fig. 1b), the ENA site is dominated by the clean maritime flow



associated with the recirculating Azores high. As the recirculating Azores high clusters are usually not associated with long-range transport, a reduced contribution to MBL aerosol from continental pollutions through FT entrainment is expected during the summertime in ENA.

**3.2 Mixing ratios of CO, O₃ and water vapor**

The mixing ratios of CO, O₃ and water vapor within the MBL are expected to be strongly influenced by entrainment of FT air in ENA. CO is a long-lived species with a lifetime of approximately 1 month (Seinfeld and Pandis, 2016), and therefore is a good indicator of long-range transported continental emissions for remote sites. At the ENA site, the influence of local emissions on trace gases and aerosol measurements is expected to be minimal after filtering of the data (section S1). The lifetime of O₃ varies

from hours in polluted urban regions (due to the high rate of photochemical reactions) to several weeks in the FT (Monks et al., 2015). Given its long lifetime in FT, O₃ may also serve as tracer for long-range transported pollutions. The local photochemical activities can be inferred from the correlation between O₃ and CO. In regions with strong local sources and sinks, O₃ and CO show a strong positive correlation during summer daytime due to photochemical reactions, but a negative correlation during winter nights due to the stronger dry deposition of O₃ than CO (Poulida et al., 1991; Chin et al., 1994). In contrast, at the ENA

site, CO and O₃ are positively correlated all year around, even in winter nighttime with low wind speed (WS) < 2 m/s (Mao and Talbot, 2004) (Fig. S3a). This suggests the variation of O₃ concentration observed at the ENA site is mainly influenced by the entrainment of FT air, in agreement with findings from previous modeling studies (Cooper et al., 2002; Voulgarakis et al., 2011). In addition, the strong anti-correlation (correlation coefficient being -0.75) of CO and O₃ with water vapor (Fig. S3b) also confirms the picture above, as water mixing ratio usually negatively correlates with the FT entrainment extent at remote marine

sites (Helmig et al., 2002). Furthermore, the seasonal variations of O₃ and CO in ENA (Fig. 2a, b) differ much from those observed at anthropogenic-influenced urban or rural sites, where ozone usually exhibits a summer peak due to strong photochemical production, while CO usually shows no clear seasonal variation (Poulida et al., 1991). In contrast, CO and O₃ in ENA show a summer minimum and spring-winter maximum, suggesting minor contributions from local emissions and in-situ photochemistry (Parrish et al., 1998; Fischer et al., 2003; Mao and Talbot, 2004). The seasonal variation of CO and O₃

concentrations is also consistent with the cluster analysis of back-trajectories, which indicates more influence from long-range transported pollution in winter-spring than in summer.

**3.3 Absorbing aerosols**

In ENA boundary layer, absorbing aerosols, including black carbon (BC), brown carbon, and dust, are likely entrained from FT

following transport from continental sources. Occasionally, air masses with very strong influences from biomass burning or dust are observed the ENA site. These episodes are excluded from the analyses presented here to focus on the long-term background variations. These episodes are identified using the aerosol optical properties (Logan et al., 2013; Logan et al., 2014; Cazorla et al., 2013), particle chemical compositions (Clarke et al., 2007), and trace gas mixing ratios (Honrath et al., 2004). Identification of the episodes and aerosol properties during these dust and BB episodes will be discussed elsewhere (Zheng et al., in prep). After

these episodes are excluded, the equivalent BC mass concentrations were estimated from PM₁ $B_{abs}$ with an assumed mass absorbing cross section of 7.5 m² g⁻¹ at 529 nm (Bond et al., 2013).





While BC particles are entrained from the FT, the seasonal variation of BC mass concentration is different from those of CO and $O_3$ (Fig. 2). As evidenced from a decreasing BC/CO ratio with increasing $P_{CB}$ (Fig. S4), such difference is due to coalescence scavenging (section 4.2) during the long-range transport and/or after entrainment into the MBL, which is a sink of BC but not of CO or $O_3$. Therefore, BC can be indicative of the overall effect of FT contribution from continental emissions and coalescence

scavenging. As shown in Fig. 2c, BC mass concentration are similar in all seasons, but show larger annual variations than CO or $O_3$ do, which can be attributed to the larger annual variations of precipitation.

### 3.4 Aerosol size distributions

### 3.4.1 Modes of aerosol size distributions

The aerosol size distribution from 60 nm to 1 μm at the ENA site typically consists of three modes (Fig. 3): an Aitken (At) mode below ~ 100 nm, an accumulation mode (Ac) which resides mainly from 100 to 300 nm, and a larger accumulation mode (LA) above ~ 300 nm. Note that due to the lower size limit of UHSAS, the Aitken mode is often not fully characterized. Therefore, its number concentration is derived by deducting fitted number concentrations of the other two modes from the total number concentration $CN$ measured by the CPC, namely $N_{At} = CN - N_{Ac} - N_{LA}$. With this definition, the derived Aitken mode

concentration also includes nucleation mode particles (i.e., $D_p < 20$ nm). However, previous studies have shown that NPF events within remote MBLs like the ENA are rare (Raes, 1995; Bates et al., 2000), therefore nucleation mode particles likely represent a small fraction of the derived Aitken mode number concentration for long term measurements (Wood et al., 2012). The Ac mode is absent in 15 % of cases (Table 2), likely due to coalescence scavenging or lack of cloud-processing (section 4). Among these three modes, aerosol number concentration is dominated by At (72.0 %) and Ac (24.9 %) modes (Fig. 3b1), while the volume

concentration is controlled by the LA (74.3 %) and Ac (25.1 %) modes (Fig. 3b2).

### 3.4.2 Seasonal variations of each mode

Different seasonal variations are observed for the three particle modes. The Ac mode exhibits higher number concentration, larger mode $D_p$, and higher occurrence in summer than in winter (Table 2). In contrast, LA mode shows opposite seasonal trends,

with the number and volume concentrations in winter almost double those in summer (Table 2). These seasonal trends are also evident in the seasonal-averaged size distributions (Fig. 4a). The monthly average concentrations and the seasonal trends of the Ac and LA modes are very consistent from 2015 to 2017, showing little annual variation (Fig. 4b). Despite the higher $N_{Ac}$ in summer, $CN$ usually peaks in spring as a result of higher Aitken mode concentration (Fig. 4b). In comparison, the monthly average $N_{At}$ and $CN$ show some minor annual variations, while their seasonal trends remain the same from 2015 to 2017 (Fig. 4b).

### 4. Governing equation of MBL aerosol number budgets and estimation of the key process terms

### 4.1 Governing equations of At, Ac, and LA mode concentrations

The mode-dependent seasonal trends indicate that different processes drive the variation of $N_{At}$, $N_{Ac}$ and $N_{LA}$. Processes that may influence ENA MBL aerosol number concentrations are entrainment of the FT particles, SSA production, NPF inside MBL,

condensation growth (COND), coagulation (COAG), in-cloud scavenging of interstitial particles by activated droplets (INT), aqueous-phase chemistry (AQ_CHEM), wet deposition, dry deposition and advection. Among these potential processes, NPF





within the MBL was shown quite rare in previous studies (Raes, 1995; Bates et al., 2000), and is neglected in the calculations of the long-term budget terms (Wood et al., 2012). Also, at remote marine sites like ENA, the influence of advection is "averaged" out for long term trends of particle concentrations. In addition, dry deposition is usually much slower compared to wet deposition for submicron particles, and thus is expected to have negligible impact on aerosol concentrations (Lewis and Schwartz, 2004;

Wood et al., 2012). Wet deposition includes both coalescence scavenging of activated droplets therefore effectively *CCN* inside clouds (COALES) and the collection of aerosol particles by falling hydrometeors below cloud (i.e., washout). For aerosols between 10 nm and 1 μm, below-cloud washout is usually much less efficient than in-cloud coalescence scavenging (Garrett et al., 2006; Seinfeld and Pandis, 2016; Wood et al., 2012), and is neglected here. Earlier study suggests that the Ac mode in MBL is formed through aqueous-phase chemistry inside activated cloud droplets (Hoppel et al., 1990). Therefore, here we treat both

Ac and LA mode particles as *CCN*, and At mode particles non-*CCN* (i.e., remain as interstitial particles inside clouds). This treatment is also supported by the strong correlation between $N_{Ac} + N_{LA}$ and *CCN* concentration at 0.2 % *ss*, representative for marine low clouds (Wood et al., 2012; Clarke and Kapustin, 2010). This strong correlation also shows little seasonal variation (Fig. S5). As $N_{Ac}$ is usually one order of magnitude higher than $N_{LA}$ (Fig. 4), $N_{Ac}$ itself can also serve as a surrogate of *CCN* concentration at the ENA site (Fig. S5). Therefore the overall governing equation for each mode of MBL aerosol can be written

as:

$$\partial_t N_{At} = \partial_t N_{At}\big|_{FT} + \partial_t N_{At}\big|_{SSA} + \partial_t N_{At}\big|_{COND} + \partial_t N_{At}\big|_{COAG} + \partial_t N_{At}\big|_{INT} \tag{1a}$$

$$\partial_t N_{Ac} = \partial_t N_{Ac}\big|_{FT} + \partial_t N_{Ac}\big|_{SSA} + \partial_t N_{Ac}\big|_{COND} + \partial_t N_{Ac}\big|_{COAG} + \partial_t N_{Ac}\big|_{AQ\_CHEM} + \partial_t N_{Ac}\big|_{COALES} \tag{1b}$$

$$\partial_t N_{LA} = \partial_t N_{LA}\big|_{FT} + \partial_t N_{LA}\big|_{SSA} + \partial_t N_{LA}\big|_{COND} + \partial_t N_{LA}\big|_{COAG} + \partial_t N_{LA}\big|_{AQ\_CHEM} + \partial_t N_{LA}\big|_{COALES} \tag{1c}$$

as depicted in Fig. 5 and discussed in details below.

### 4.2 Key aerosol sources and sinks

**SSA**

The MBL aerosol concentration change rate due to SSA production flux, $\partial_t N\big|_{SSA}$, can be expressed as (de Leeuw et al., 2011; Wood et al., 2012):

$$\partial_t N\big|_{SSA} = \frac{3.84 \times 10^{-6}\ \mathrm{WS}^{3.41} F_{SSA}}{H_{MBL}} = \frac{3.84 \times 10^{-6}\ \mathrm{WS}^{3.41}}{H_{MBL}} \int_{D_p} f_{SSA}(lnD_p)\, dlnD_p \tag{2}$$

where $3.84 \times 10^{-6}\ \mathrm{WS}^{3.41}$ is the white cap fraction on the sea surface (Monahan et al., 1986) with WS in unit of m s⁻¹, $F_{SSA}$ is the total SSA number production flux per white cap area in unit of m⁻² s⁻¹, $H_{MBL}$ is the MBL height in m, and $f_{SSA}(lnD_p)$ is the lognormal number size distribution of SSA production flux curve. Thus, WS is the most important parameter in estimating total SSA contributions, while the detailed size distribution could differ with the $f_{SSA}(lnD_p)$ used (Gong, 2003; Lewis and Schwartz,

2004; Clarke et al., 2006; Grythe et al., 2014).

**Coalescence scavenging**

The rate of coalescence scavenging of cloud droplets is given by (Wood, 2006; Wood et al., 2012)



$$\partial_t N_d\big|_{COALES} = E\big|_{COALES}\, N_d = -N_d K P_{CB} h H_{MBL}^{-1}$$

(3)

where $E|_X$ represents $(\partial_t N|_X)/N$, namely the percentage processing efficiency of process X. $N_d$ is cloud droplet number concentration which can be assumed to be the same as $CCN$, or $N_{Ac} + N_{LA}$ (section 3.4.2); $K$ is a constant of 2.25 m²/kg; while $h\,H_{MBL}^{-1}$ represents the in-cloud volume fraction of MBL aerosols (MÅrtensson et al., 2010). Note that by letting the precipitation

rate at cloud base, $P_{CB}$, as 0 when there is no precipitation, the precipitation time fraction is already included in Eq. 3.

**In-cloud scavenging of interstitial particles by activated droplets**

Inside clouds, interstitial particles are scavenged when coagulating with activated cloud droplets. This process directly reduces At mode particle number concentration, while also indirectly reducing $CCN$ (i.e., Ac and LA modes) number concentration by

removing particles that could otherwise grow and become $CCN$ (Pierce et al., 2015). The rate of scavenging scales with the probability that the particles are inside clouds, $f_{cloud}$. Here $f_{cloud}$ is defined as:

$$f_{cloud} = p_{cloud}\, h\, H_{MBL}^{-1}$$

where $p_{cloud}$ is the probability that MBL cloud is encountered and is approximated by the in-cloud time fraction (Table 1), while $h\,H_{MBL}^{-1}$ is again indicative of the volume fraction of MBL aerosol particles inside the clouds (MÅrtensson et al., 2010).

As At mode particles are treated as non-$CCN$ and remain as interstitial particles inside the clouds, the rate of the scavenging can be estimated by (Pierce et al., 2015):

$$\partial_t N_{At}\big|_{INT} = -f_{cloud} K_{int,d} N_{At} N_d, \text{ namely } E\big|_{INT} = -f_{cloud} K_{int,d} N_d$$

(4)

where $N_d$ is number concentration of cloud droplets assumed to be the sum of $N_{Ac}$ and $N_{LA}$ (section 3.4.2), $K_{int,d}$ is the coagulation coefficient between $D_{p,int}$ and $D_{p,d}$, where $D_{p,int}$ and $D_{p,d}$ represent the diameter of interstitial particles and cloud

droplets, respectively. $D_{p,d}$ is assumed to be 10 μm (Pierce et al., 2015), while $D_{p,int}$ is assumed to be the corresponding wet diameter of $D_{pg,At}$ under a supersaturation of 0.12 % (Korolev and Mazin, 2003), where $D_{pg,At}$ is the geometric mean dry diameter of At mode. The maximum supersaturation near the cloud base where $CCN$ activation occurs is typically 0.2 % for marine low clouds (Wood et al., 2012; Clarke and Kapustin, 2010). However, the suspersaturation is usually lower above the cloud base where most of the interstitial scavenging occurs. Here we assumed the in-cloud $ss$ of 0.12 % based on the work of

Korolev and Mazin (2003). Assuming the At mode have a minimum $D_p$ of 23 nm (Pandis et al., 1994), the $D_{pg,At}$ is estimated as 48nm, and the corresponding wet particle diameter inside clouds, $D_{p,int}$, is around 190 nm. Sensitivity of the interstitial scavenging rate to these parameters is discussed in section 6.3.

**Aqueous-phase chemistry**

Aqueous-phase reactions inside activated cloud droplets do not change total particle number concentration. On the other hand, it can efficiently add mass to $CCN$ and grow them into larger diameters when cloud droplets evaporate following the reactions. Therefore, the only influence of AQ_CHEM on number size distribution considered here is the growth of Ac mode particles into the LA mode size ranges. The magnitude of the influence depends on $f_{cloud}$, liquid water content, precursor concentrations, and the radiation which would influence oxidant concentrations (MÅrtensson et al., 2010).





**Condensation growth**

While condensation does not change the total particle number concentration, it grows the particles and therefore changes the number distribution among different modes (Seinfeld and Pandis, 2016). In this sense it functions similarly to aqueous-phase reactions, with the difference being that condensation acts on particles of all sizes while aqueous-phase reactions influence only

CCN. The rate of a smaller mode A growing into a larger mode B through condensation can be estimated as (Pandis et al., 1994):

$$\partial_t N_B\big|_{COND} = -\partial_t N_A\big|_{COND} = (1 - f_{cloud}) J_V(A) / \Delta V_A$$

(5)

where $\Delta V_A$ (in $\mu m^3$) is the volume difference between a particle with the minimum $D_p$ of mode B, and a particle with the volume average of mode A, namely:

$$\Delta V_A = \frac{\pi}{6} D_{p2}{}^3 - \frac{\int_{D_{p1}}^{D_{p2}} \frac{\pi}{6} D_p{}^3 n(D_p) dD_p}{\int_{D_{p1}}^{D_{p2}} n(D_p) dD_p}$$

where the integrals are calculated from the binned aerosol size distribution using the binned-simplification described in Pandis et al. (1994).

$J_V(A)$ is the volume condensation rate of mode A in $\mu m^3$ $m^{-3}$ $s^{-1}$, which can be estimated as (Seinfeld and Pandis, 2016):

$$J_V(A) = K_{COND}(A) \frac{P}{RT\rho_p} m_i (v_i - v_{eq})$$

where $K_{COND}(A)$ is the condensation rate constant of mode A in $s^{-1}$, $R$ is the gas constant of 8.314 J $mol^{-1}$, $T$ is temperature in K, $P$ is the atmospheric pressure being $1.013 \times 10^5$ Pa, $\rho_p$ is the aerosol density assumed to be $1 \times 10^{-12}$ g $\mu m^{-3}$, $v_i$ and $v_{eq}$ are the volume mixing ratio of condensate in the bulk gas-phase and at the aerosol surface, and $m_i$ is the molar mass of condensate. Here we assume that the condensate is $H_2SO_4$ and thus $m_i = 98$ g $mol^{-1}$, and $v_{eq}$ is 0 (Pandis et al., 1994). Annual mean $v_i$ is assumed to be 1.0 ppt (Pandis et al., 1994), while being 1.4, 1.3, 1.1 and 0.2 ppt in spring, summer, fall and winter, respectively. This

seasonal variation in $v_i$ is based on the monthly dimethyl sulfide (DMS) fluxes (assumed to be 7.0, 5.4, 2.9 and 1.0 $\mu mol$ $m^{-2}$ $day^{-1}$ in spring, summer, fall and winter, respectively) given in previous studies in the North Atlantic Ocean (Tarrasón et al., 1995), and the proposed dependence of $H_2SO_4$ on DMS flux at the observed fluxes ranges (Pandis et al., 1994; Russell et al., 1994). $K_{COND}(A)$ can be estimated by (Seinfeld and Pandis, 2016):

$$K_{COND}(A) = 2 \times 10^{-4} \pi D \int_{D_{p1}}^{D_{p2}} g_f D_p f(Kn, \alpha) n(D_p) dD_p$$

where $10^{-4}$ is the unit converter of $\mu m$ $cm^{-1}$, $D$ is the gas diffusivity of condensate in air equaling 0.1 $cm^2$ $s^{-1}$, $g_f$ is the aerosol hygroscopic growth factor at ambient RH, $D_p$ is the dry aerosol diameter in $\mu m$, $n(D_p)$ is the number size distribution of mode A in $\mu m^{-1}$ $cm^{-3}$, $D_{p1}$ and $D_{p2}$ are the diameter boundaries of mode A and defined as the corresponding mode gap $D_p$ in Table 2 here, and $Kn$ is the Knudsen number given by $2\lambda_{mfp} (g_f D_p)^{-1}$, where $\lambda_{mfp}$ is the air mean free path. At the ENA site, observed ambient RH show mild diurnal variation of 75 % ± 10 %. Accordingly, the hygroscopic growth factor, $g_f$, is assumed to be 1.3, based on

the Hygroscopic Tandem Differential Mobility Analyser measurements of At and Ac mode particles at the ENA site. The increased particle surface area due to hygroscopic growth leads to a factor of ~1.7 increase in the estimated $K_{COND}$ compared with





that under dry conditions. The term $f(Kn, \alpha)$ is the correction due to non-continuum effects (scaled by $Kn$) and imperfect surface accommodation (scaled by the mass accommodation coefficient α) estimated by the Fuchs-Sutugin approach as (Seinfeld and Pandis, 2016):

$$f(Kn,\alpha) = \frac{0.75\alpha(1+Kn)}{Kn^2 + (1+0.283\alpha)Kn + 0.75\alpha}$$

where α is assumed to be 0.02 for $H_2SO_4$ (Pandis et al., 1994).

**Coagulation**

Unlike condensation, coagulation does not change the total mass concentration, but reduces aerosol number concentrations. The intra-modal coagulation of particles in a smaller mode A (e.g., At mode) serves as both a source of particles in a larger mode B

(corresponding rate denoted as $J_{AA\rightarrow B}$ hereinafter) and a sink for particles of mode A. Given the typical aerosol size distribution observed at the ENA site, intra-modal coagulation of mode A particles is usually negligible when compared to inter-modal coagulation between mode A and another mode B with a different size range (corresponding rate denoted as $J_{AB}$ hereinafter) (Dal Maso et al., 2002). Therefore we focus on the intra-modal coagulation as particle source of a larger mode, and inter-modal coagulation as a particle sink. The corresponding rate, $J_{AA\rightarrow B}$ and $J_{AB}$, are respectively estimated as:

$$J_{AA\rightarrow B} = 0.5 \int_{D_{p\min,A}}^{D_{p\max,A}} \int_{D_{pc}}^{D_{p\max,A}} K_{12} n(D_{p1}) n(D_{p2}) dD_{p1} dD_{p2}$$

$$J_{AB} = \int_{D_{p\min,B}}^{D_{p\max,B}} \int_{D_{p\min,A}}^{D_{p\max,A}} K_{12} n(D_{p1}) n(D_{p2}) dD_{p1} dD_{p2}$$

where $K_{12}$ is the coagulation coefficient between two particles with diameters of $g_fD_{p1}$ and $g_fD_{p2}$, respectively, and is calculated using the Fuchs form (Seinfeld and Pandis, 2016). Similarly as in the estimation of $K_{COND}$, the growth factor under ambient RH, $g_f$, is assumed to be 1.3. This increase in particle diameter results in a ~20 % decrease in estimated $K_{12}$. $D_{p\min, A}$ and $D_{p\max, A}$ are

the boundary diameter of mode A (defined as the corresponding mode gap $D_p$ in Table 2 here), while $D_{p\min, B}$ and $D_{p\max, B}$ is defined similarly. $D_{pc}$ is defined by

$$D_{pc}^3 = D_{pB, \min}^3 - D_{p1}^3$$

The coagulation loss rate of $N_A$ is thus:

$$\partial_t N_A \big|_{COAG} = -(1 - f_{cloud}) \sum_B J_{AB}$$

(6a)

while the coagulation production rate of $N_B$ is:

$$\partial_t N_B \big|_{COAG} = (1 - f_{cloud}) J_{AA\rightarrow B}$$

(6b)





### 4.3 Estimated rate of the potential key processes

The terms in the governing equations 1(a-c) due to condensation, coagulation, scavenging of interstitial aerosol, and coalescence scavenging of $CCN$ are estimated using and equations described above and the size distribution parameters listed in Table 2. The values are listed in Table 3. The discussions of these estimates follow in section 5 and 6.

## 5 Controlling processes of larger accumulation mode

Potential influencing processes of the LA mode number concentration include:

$$\partial_t N_{LA} = \partial_t N_{LA}|_{FT} + \partial_t N_{LA}|_{SSA} + \partial_t N_{LA}|_{COND} + \partial_t N_{LA}|_{COAG} + \partial_t N_{LA}|_{AQ\_CHEM} + \partial_t N_{LA}|_{COALES}$$

(1c)

Among these processes, SSA is expected to be the dominant source of $N_{LA}$ in MBL, as suggested by the strong correlation between $N_{LA}$ and WS (Fig. 6b), a key parameter of SSA production flux (section 4). Aqueous-phase reactions could also produce
"droplet mode" particles in the LA mode size range (Pandis et al., 1990; Meng and Seinfeld, 1994). However, if aqueous-phase reactions present a major source, we would expect the volume size distribution exhibits a mode $D_p$ of 0.6~0.8 μm, corresponding to the size ranges that have the largest access to cloud water (Pandis et al., 1990; Seinfeld and Pandis, 2016). In contrast, the volume size distribution indicates that the LA mode is actually the leading edge of a larger mode with peak $D_p$ in the super-micron range (Fig. 3b2). This is also supported by the strong correlation between $V_{LA}$ and PM$_c$ $B_{sca}$ (Fig. 6c), which is a surrogate
for the coarse mode volume concentration (section 2.2). Therefore, we expect LA particles, like supermicron particles, are dominated by SSA in remote MBL (Campuzano-Jost et al., 2003). The contribution of aqueous-phase reactions to the LA mode number concentration is likely minor, and is neglected from the governing equation Eq. (1c) in following analysis.

Given the large sizes of LA mode particles, we do not expect any significant FT sources. The lack of correlation between $N_{LA}$
and BC mass concentration also suggests a low concentration of LA mode particles in long-range transported continental pollution plumes. Here we assume the concentration of LA mode particles is negligible when compared to that in the boundary layer. In such case, the entrainment of FT air dilutes the MBL LA particles, serving as a sink rather than a source. At a typical entrainment velocity, $\omega_e$, of 3.5 mm s$^{-1}$ (Mohrmann et al., 2018; Wood and Bretherton, 2004), the maximum dilution rate, $-E_{LA}|_{FT}$, equaling $\omega_e H_{MBL}^{-1}$ (Mohrmann et al., 2018), reaches ~ 20 % per day. That is comparable to coalescence scavenging, making the
FT dilution an important sink of $N_{LA}$. The sensitivity of the rate to entrainment velocity is discussed at the end of this section.

The terms of intra-modal coagulation ($\partial_t N_{LA}|_{COAG}$) and condensation ($\partial_t N_{LA}|_{COND}$) from Ac mode are estimated as 0.02 and 0.6 cm$^{-3}$ day$^{-1}$, respectively (Table 3). Both of these two processes are too slow to exert significant influences during the typical aerosol lifetime of 7~10 days. The governing equation of $N_{LA}$ (Eq. 1c) can therefore be simplified into:

$$\partial_t N_{LA} = \partial_t N_{LA}|_{FT} + \partial_t N_{LA}|_{SSA} + \partial_t N_{LA}|_{COALES}$$

(7)

The seasonal variation of LA mode number concentration is a result of the balance among the three processes (Fig. 6). Production flux of SSA is proportional to WS$^{3.41}$ $H_{MBL}^{-1}$ (Eq. 2), coalescence scavenging efficiency is $K P_{CB} h H_{MBL}^{-1}$ (Eq. 3), and FT dilution efficiency is estimated as $\omega_e H_{MBL}^{-1}$. Among these three terms, FT entrainment term (Fig. 6e) shows little seasonal
variation. In comparison, both coalescence scavenging (Fig. 6d) and SSA production (Fig. 6f) terms are lower in summer while





higher in winter, while the SSA production has a larger variation. The value of $N_{LA}$ under the quasi-steady-state, (i.e. when $\partial_t N_{LA}$ = 0) can be scaled using the three terms for each seasons. The scaled steady-state $N_{LA}$ (red markers in Fig. 6g) successfully produces the observed seasonal pattern trend of $N_{LA}$ (boxplots and black lines Fig. 6g). Varying the value of the assumed entrainment velocity within the typical range of 2~5 mm s$^{-1}$ does not affect the overall seasonal trend of the scaled $N_{LA}$.

**6. Controlling processes of Aitken - Accumulation mode**

**6.1 Contributions of SSA to Aitken and Accumulation modes**

Unlike $N_{LA}$, $N_{At}$ and $N_{Ac}$ are either independent of the WS, or decrease with increasing WS (Fig. 7), indicating relatively minor contributions from SSA to At and Ac modes. Both $N_{At}$ and $N_{Ac}$ increases monotonically with BC mass concentration (Fig. 7), suggesting At and Ac modes are dominated by the long-range transported anthropogenic particles.

A semi-quantitative estimation of SSA contribution also supports the above conclusion. Assuming all LA mode particles are from SSA, by combining $N_{LA}$ and an established size distribution of SSA production flux, one can estimate the upper limit of the SSA contribution to At and Ac modes (Fig. 8). For simplification, here we use number concentration of particles with $D_p$ in the range from 400 to 1000 nm, $N_{400}$, to represent the observed SSA number concentration in the same $D_p$ range. SSA larger than

~100 nm are $CCN$ under $ss$ of 0.1 % (Petters and Kreidenweis, 2007), while the measured $CN$ has a cut-off diameter of roughly 10 nm. The contribution of SSA to $CCN$(0.1 %) and $CN_{SSA}$ can therefore be estimated by:

$$CCN(0.1\%)_{SSA} = k_{CCN} N_{400} \frac{\int_{\ln 100}^{\ln 1000} f_{SSA}(\ln D_p) d\ln D_p}{\int_{\ln 400}^{\ln 1000} f_{SSA}(\ln D_p) d\ln D_p}$$

$$CN_{SSA} = k_{CN} CCN(0.1\%)_{SSA} = CCN(0.1\%)_{SSA}\left(1 + k_{INT}\frac{\int_{\ln 10}^{\ln 100} f_{SSA}(\ln D_p) d\ln D_p}{\int_{\ln 100}^{\ln 1000} f_{SSA}(\ln D_p) d\ln D_p}\right)$$

where $k_{CCN}$ and $k_{INT}$ are factors that account for the size dependence of removal rate (see derivations in SI S3). The estimated

$k_{CCN}$ is around 1, while $k_{INT}$ can vary from 1.7 to 4.4 as the removal efficiency is higher for $CCN$ than non-$CCN$ (Table 4, Fig. 9).

Here we used four published $f_{SSA}(\ln D_p)$ schemes (Gong, 2003; Lewis and Schwartz, 2004; Clarke et al., 2006; Grythe et al., 2014) to calculate the contribution of SSA to observed $CCN$(0.1 %) and $CN$ (Fig. 8a). The initial calculation neglects the size dependence of the particle removal rate, therefore the results represent lower limits of the contributions (Fig. 8b). This approach

essentially assumes that the shape of SSA size distribution in MBL is the same as that of SSA flux. Even for these lower limit estimates, $CCN$(0.1 %)$_{SSA}$ and $CN_{SSA}$ calculated using $f_{SSA}(\ln D_p)$ from Gong et al. (2003) and Clark et al. (2006) exceed the observed total $CCN$(0.1 %) and $CN$ for a substantial fraction of the data, suggesting that these two $f_{SSA}(\ln D_p)$ functions result in overestimation of SSA contributions over the 10-400 nm size range at the ENA site. This may be partially due to the parameter dependencies of sea surface temperatures, etc. (Gantt and Meskhidze, 2013; Gantt et al., 2015; Quinn et al., 2015), which are not

considered here. The value of $CCN$(0.1 %)$_{SSA}$ and $CN_{SSA}$ are therefore estimated as the averages of predictions based on flux size distributions reported by Grythe et al (2014) and Lewis and Schwartz (2004), with $k_{CCN}$ and $k_{INT}$ taken into consideration. The corresponding mean fractions of $CCN$(0.1 %)$_{SSA}$ and $CN_{SSA}$ in observed $CCN$(0.1 %) and $CN$ are 24 % and 11 %, respectively. The estimated $CCN$(0.1 %)$_{SSA}$ fraction is in consistent with a recent study that shows that the SSA contribution to $CCN$ is smaller




than 30 % globally (Quinn et al., 2017). In that study, the size distribution of SSA was derived by fitting aerosol size distribution. The SSA number concentration is therefore $N_{LA}$ following the approach, and it represents 19 % of $CCN(0.1\%)$.

Based on above estimation, the contributions of SSA to $N_{Ac}$ and $N_{At}$ are estimated as:

$$f_{Ac,SSA} = (CCN(0.1\%)_{SSA} - N_{LA})/N_{Ac}$$

$$f_{At,SSA} = (CN_{SSA} - CCN(0.1\%)_{SSA})/N_{At}$$

and the corresponding annual mean contributions to $N_{Ac}$ and $N_{At}$ are 21 % and 10 %, respectively (Table 4).

### 6.2 Controlling processes of Accumulation mode

10  As shown in section 6.1, the contribution of SSA to Ac mode is likely substantial (annual average ~21 %, Table 4). For the Ac mode, both intra-modal and inter-modal coagulations are much slower than coalescence scavenging (Table 3), therefore can be neglected from the governing equation of $N_{Ac}$. On the other hand, condensation growth of At mode particles may represent a substantial source of Ac mode. AQ_CHEM reduces $N_{Ac}$ by growing particles into LA size range. As discussed in section 5, such process only makes a minor contribution to $N_{LA}$, Therefore the impact of AQ_CHEM on $N_{Ac}$ is expected to be negligible given

15  $N_{Ac}$ is about one order of magnitude higher than $N_{LA}$. The governing equation of $N_{Ac}$ (Eq. 3b) can be simplified into:

$$\partial_t N_{Ac} = \partial_t N_{Ac}\big|_{FT} + \partial_t N_{Ac}\big|_{SSA} + \partial_t N_{Ac}\big|_{COND} + \partial_t N_{Ac}\big|_{COALES}$$

(8)

The estimated values of $\partial_t N_{Ac}|_{COND}$ and $\partial_t N_{Ac}|_{COALES}$ are listed in Table 3. Coalescence scavenging is the only sink of Ac mode particles among the four main processes, while the other three are sources. Under steady-state conditions, ($\partial_t N_{Ac} = 0$), we have:

$$f_{SSA,\,Ac} = \frac{\partial_t N_{Ac}\big|_{SSA}}{\partial_t N_{Ac}\big|_{FT} + \partial_t N_{Ac}\big|_{COND} + \partial_t N_{Ac}\big|_{SSA}} = -\frac{\partial_t N_{Ac}\big|_{SSA}}{\partial_t N_{Ac}\big|_{COALES}}$$

20  Namely

$$\partial_t N_{Ac}\big|_{SSA} = -f_{SSA,\,Ac}\,\partial_t N_{Ac}\big|_{COALES}$$

The contribution due to the entrainment of FT air, $\partial_t N_{Ac}|_{FT}$, can be estimated as a residual using the $f_{SSA,\,Ac}$ value derived in the previous section (Table 4):

$$\partial_t N_{Ac}\big|_{FT} = -(1 - f_{SSA,\,Ac})\,\partial_t N_{Ac}\big|_{COALES} - E_{Ac}\big|_{COND}$$

(9)

The normalized rates of different processes are compared in Fig. 9a. The derived $\partial_t N_{Ac}|_{FT}$ is stronger in winter-spring while lower in summer-fall, in general agreement with the seasonal trends of observed CO and BC (Fig. 9b), consistent with the picture that anthropogenic emissions represent the main source of entrained FT Ac mode particles. The agreement also suggests that the above analysis captures the major seasonal variation of the contribution of FT entrainment to Ac mode particles.





Based on the first-order estimates shown in Fig. 9, we can see that on an annual basis, entrainment from FT represents the major source of $N_{Ac}$, followed by condensation growth of At mode particles and SSA production. However, the relative importance of these three sources shows substantial seasonal variations. Contribution from SSA production is the lowest in summer (12 %), and the highest during winter (31 %; Table 4), a result of strong seasonal variation of surface wind speed. In contrast, condensation is negligible in winter due to the substantial lower DMS emissions and thus $H_2SO_4$ concentrations (section 4.2). In summer and fall, however, contribution from condensation growth (60 % in summer and 42 % in fall) exceeded that from FT entrainment, and became the dominate source of the Ac mode. This variation in relative importance despite similar $E_{Ac}|_{COND}$ in spring to fall (Fig. 9a) is mainly due to the large seasonal variation in FT entrainment efficiencies. Because sulfuric acid is expected a production of DMS oxidation, the large contribution of condensation growth to $N_{Ac}$ suggests that ocean biological activity may have a substantial influence on MBL $CCN$ populations through the production of DMS by phytoplankton. The estimated contribution from condensation is consistent with observations of individual aerosol particles in western Atlantic (Sanchez et al., 2018), but is substantially higher than that simulated over the remote Southern Hemisphere oceans during summertime (Hannele et al., 2008). Such difference is likely due to the much higher DMS flux in ENA (Kettle et al., 1999), the difference between observed and model-simulated aerosol size distributions, etc..

**6.3 Controlling processes of Aitken mode**

The governing equation of $N_{At}$ is:

$$\partial_t N_{At} = \partial_t N_{At}\big|_{FT} + \partial_t N_{At}\big|_{SSA} + \partial_t N_{At}\big|_{COND} + \partial_t N_{At}\big|_{COAG} + \partial_t N_{At}\big|_{INT} \tag{3a}$$

Following the same approach in Section 6.2, we have:

$$\partial_t N_{At}\big|_{FT} = -(1 - f_{SSA, At})(\partial_t N_{At}\big|_{COAG} + \partial_t N_{At}\big|_{COND} + \partial_t N_{At}\big|_{INT}) \tag{10}$$

Contribution of SSA to the At mode is even smaller than it is to the Ac mode, and is estimated to be no larger than 10 % (Table 4). As a result, the entrainment of FT At mode particles represents the dominant source (Fig. 9a). $\partial_t N_{At}|_{FT}$ is higher in spring-summer while lower in fall-winter, and such seasonal variation is somewhat different from those of CO mixing ratio and BC mass concentrations. These differences may be partially due to stronger new particles formation from biogenic precursors in the FT during spring and summer seasons (Sanchez et al., 2018). The strength of new particle formation is not correlated with CO or BC concentrations, which are tracers for anthropogenic emissions.

On an annual basis, inter-modal coagulation is the major (55 %) sink of $N_{At}$ (Fig. 9), followed by condensation growth (28 %) and interstitial scavenging (16 %). While it is less important when compared to inter-modal coagulation, interstitial scavenging is substantial and cannot be neglected. This is consistent with the finding of Pierce et al. (2015). The overall removal efficiency of $N_{At}$ (~10 % day$^{-1}$) is substantially lower than those of $N_{Ac}$ and $N_{LA}$, which corresponds to a longer lifetime (~ 10 days) for At mode particles in MBL. The $N_{At}$ removal efficiency is higher in summer and lower in winter, which is opposite to that of Ac and LA modes. This is partially due to the less efficient removal of At mode particles by coagulation and interstitial scavenging in winter, as a result of lower $N_{Ac}$ and therefore droplet number concentrations. In addition, the low DMS fluxes during winter (section 4.2) leads to substantially weakened condensation growth of At mode particle into Ac mode size ranges, which also contribute to the lower overall removal efficiency in winter. Relative importance of these three removal processes is quite





consistent in spring to fall, with contribution from coagulation, condensation and interstitial scavenging being around 51 %, 33 % and 16 %, respectively. In winter, condensation is a negligible (7 %) removal processes, while contribution of coagulation dominated (71 %), with the remaining 22 % due to interstitial scavenging.

We note there may be uncertainties in the above estimates, especially the rate of interstitial scavenging, which depends on Aitken mode size distribution, the super-saturation inside clouds, as well as the effective cloud droplet diameters. Based on the assumed baseline conditions (effective cloud droplet diameters of 10 um, average dry interstitial aerosols of 48 nm, and average $ss$ of 0.12 %; see section 4.1), relative sensitivities of the $K_{int, d}$ are 10 % / μm, -5 % / nm, and -8 % / % with respect to changes in droplet diameter, dry interstitial aerosol diameter, and average $ss$, respectively. For average cloud droplet diameter at 15 μm, geometrical

mean Aitken mode diameter of 45 nm and average $ss$ of 0.1 %, a condition that is more favorable for intestinal scavenging, corresponding interstitial scavenging rate would increases by a factor of 1.8.

Given the low contribution of SSA to the At mode particles, the governing equation for Aitken mode, Eq. (3b) can be simplified into:

$$\partial_t N_{At} = \partial_t N_{At}\big|_{FT} + \partial_t N_{At}\big|_{COAG} + \partial_t N_{At}\big|_{COND} + \partial_t N_{At}\big|_{INT} \tag{11}$$

### 7. Conclusion

We examine the seasonal variations of aerosol properties, trace gas mixing ratios, and meteorological parameters measured at the ARM ENA site on Graciosa Island from 2015 to 2017. Aerosol size distributions from 60 nm to 1 μm typically consist of three

modes: At (< 100 nm), Ac (100 to ~300 nm) and LA (> 300 nm). Observed $CCN$ number concentrations are in general agreement with the sum of $N_{Ac}$ and $N_{LA}$. The particle number concentration and mode diameter of the three modes exhibits different seasonal variations, suggesting that they are controlled by different processes.

LA mode particles are dominated by SSA. The major sinks of $N_{LA}$ are coalescence scavenging and dilution by entrained FT air.

$N_{LA}$ is higher in winter and lower in summer. The higher $N_{LA}$ during winter is attributed to strong SSA production flux due to high wind speed, which prevails over an increase in coalescence scavenging. The seasonal variation of steady-state $N_{LA}$ is derived from scaling the rates of major processes, and the result agrees well with the observation.

In comparison, SSA represents a relatively minor fraction of $N_{Ac}$ and $N_{At}$, with estimated annual mean contributions being 21 %

and no larger than 10 %, respectively. For $N_{Ac}$, the other sources are entrained FT Ac mode particles and condensation growth of Aitken mode particle inside MBL, while the major sink is coalescence scavenging. The derived FT contribution to $N_{Ac}$ generally follows the seasonal trends of CO and BC, namely higher in spring-winter and lower in summer, consistent with the picture that anthropogenic emissions represent the main source of entrained FT Ac mode particles. While entrainment from FT is the major source on the annual basis, the relative importance of the different sources varies strongly with the season. In summer and fall,

condensation growth of At mode may become the dominant source, contributing 60 % and 42 % of the Ac mode particles in MBL. In winter, SSA contributes to ~ 31 % of the Ac mode, surpassing the contribution due to condensation growth. This is due to a combination of strong surface wind speed and lower DMS emissions during winter season.





For $N_{At}$, entrainment from FT is expected to be the dominant source, and coagulation represents the major sink. The derived FT contribution to $N_{At}$ is higher in spring-summer and lower in fall-winter, possibly due to stronger NPF from biogenic precursors during spring and summer seasons (Tarrasón et al., 1995). The relative importance of NPF and long-range transported

continental emissions to FT Aitken and nucleation mode particles, and the subsequent contribution to MBL *CCN* population will be examined in future studies. On an annual basis, 52 %, 32 % and 16 % of $N_{At}$ are removed by inter-modal coagulation, condensation growth and interstitial scavenging, respectively. Relative importance of these three removal processes in spring to fall is quite similar with that annually. In winter, however, condensation becomes a negligible (7 %) removal process due to the low DMS fluxes, while contribution of coagulation increased to 71 %.

Based on the above results, the processes that control the concentration of the different particle modes are summarized in Fig. 10. These results suggest particles entrained from the free troposphere represent the major source of *CCN* in the marine boundary layer. Some of the entrained particles directly contribute to the Ac mode population in the MBL, and are sufficiently large to serve as *CCN*. In addition, Aitken mode particles in the free troposphere, which are attributed to NPF and long-range transported

continental emissions, can grow and form *CCN* after their entrainment into the MBL. Our calculation suggests that this represents a significant source of MBL *CCN* all year, with the highest contribution of nearly 60 % during summer seasons. As the growth of Aitken mode particles to *CCN* size is to a large degree the result of the condensation of sulfuric acid, a product of DMS oxidation, this suggests that ocean ecosystems may have a substantial influence on MBL *CCN* population in ENA through emission of DMS.

**Acknowledgements**

The research was conducted with funding from the Atmospheric System Research (ASR) and Atmospheric Radiation Measurement (ARM) programs (Office of Biological and Environmental Research of US DOE, under contract DE-AC02-98CH10886). We acknowledge the ARM Climate Research Facility, a user facility of the United States Department of Energy (US DOE), Office of Science, sponsored by the Office of Biological and Environmental Research. Robert Wood acknowledges

funding from ASR grant DE-SC0013489 (ENA Site Science).

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





**Figures and Tables**

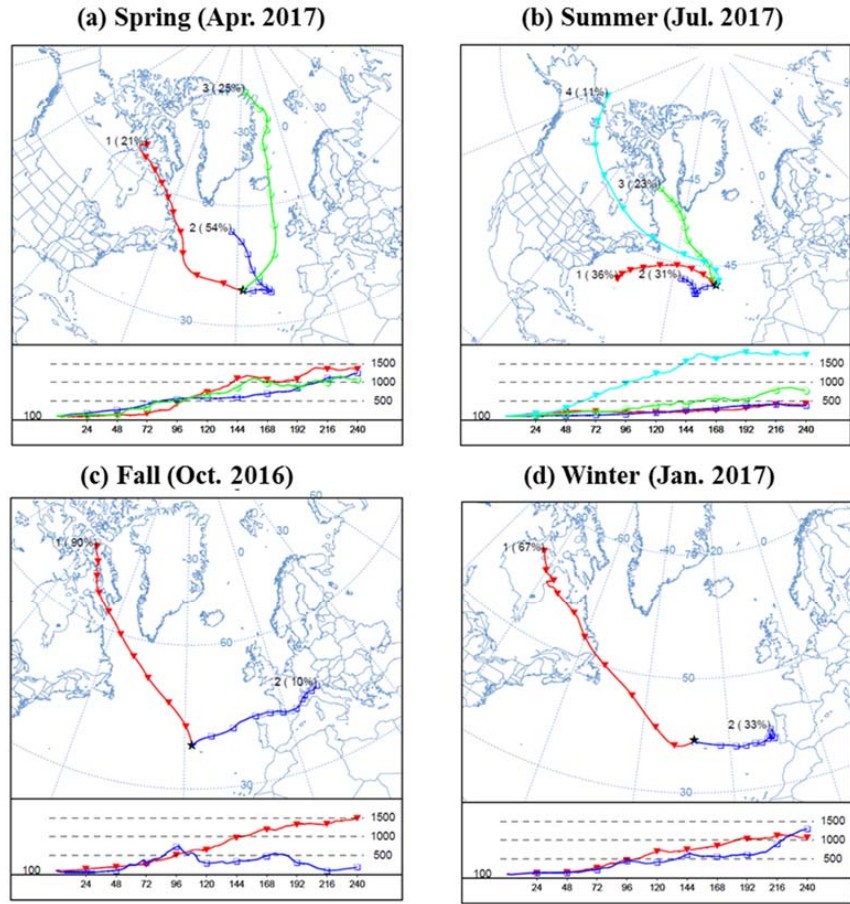

**Figure 1. Cluster analysis of 10-day back-trajectories arriving at 100 m above the ENA site in different seasons.** The analysis was conducted using the Hybrid Single-Particle Lagrangian Integrated Trajectory (HYSPLIT) 4 model (Stein et al., 2015). The 10-day back trajectories were simulated with a time step of 6 hours using National Centers for Environmental Prediction (NCEP) Global Data Assimilation System (GDAS) meteorological data as input. A cluster analysis of these trajectories was then performed, and for each season, the solution that captures most of the variance (e.g., Abdalmogith and Harrison (2005)) and with less than 5 identified clusters is chosen. The average trajectories of the clusters are represented by different colors, and the associated numbers denoted the arbitrarily given cluster ID and the occurrence percentages of this cluster. For example, number of 1 (90 %) beside the red trajectories indicated that the No. 1 cluster has an average trajectory shown by the red lines, and at 90 % times the air masses arriving at the ENA site belong to this cluster.





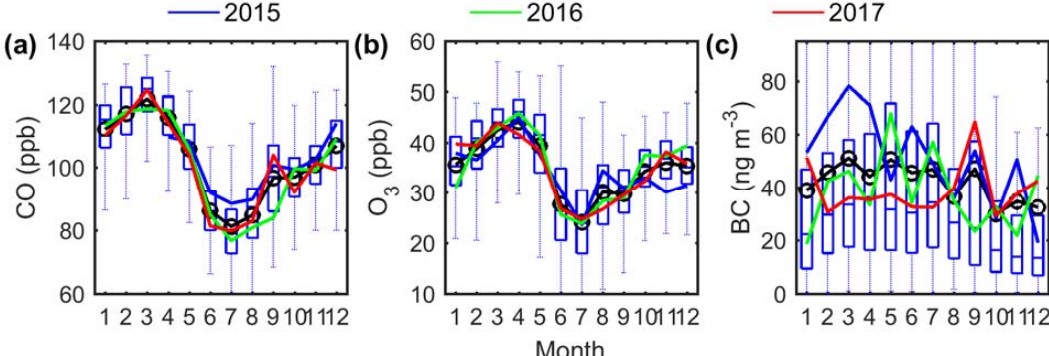

**Figure 2. Seasonal variations of** (a) CO mixing ratio, (b) $O_3$ mixing ratio, and (c) BC concentration at the ENA site. The blue, green, and red lines represent the monthly average for the year 2015, 2016, and 2017, respectively. The whiskers and boxes show the 90th, 75th, median, 25th and 10th percentiles, and the black circle and line represent the mean value of each month for the entire three years.



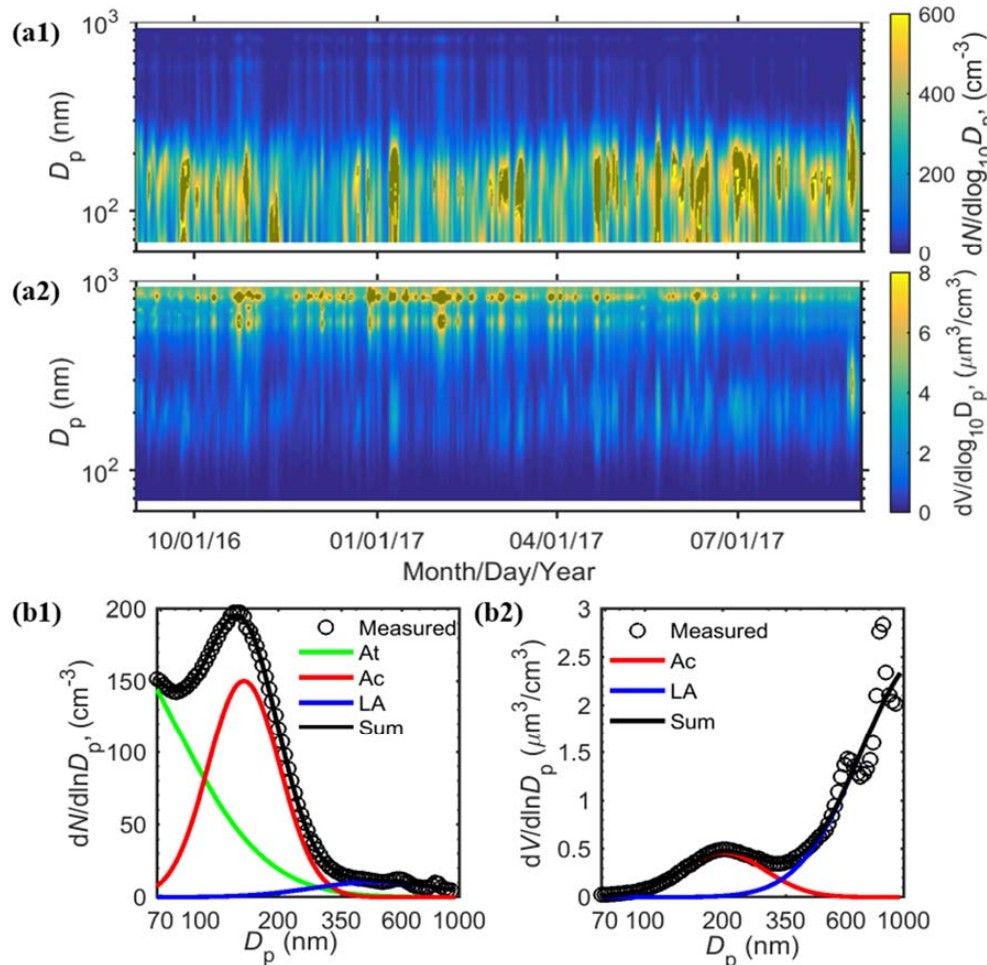

**Figure 3. Representative aerosol size distribution measured at the ENA site.** Time series of the (a1) number and (a2) volume size distributions during the study period from Sept. 2016 to Aug. 2017, and the fitted lognormal modes of (b1) number and (b2) volume distributions averaged over the one year period. The fluctuations at ~600 nm (also seen in Fig. 4) are considered as instrumental artefacts.



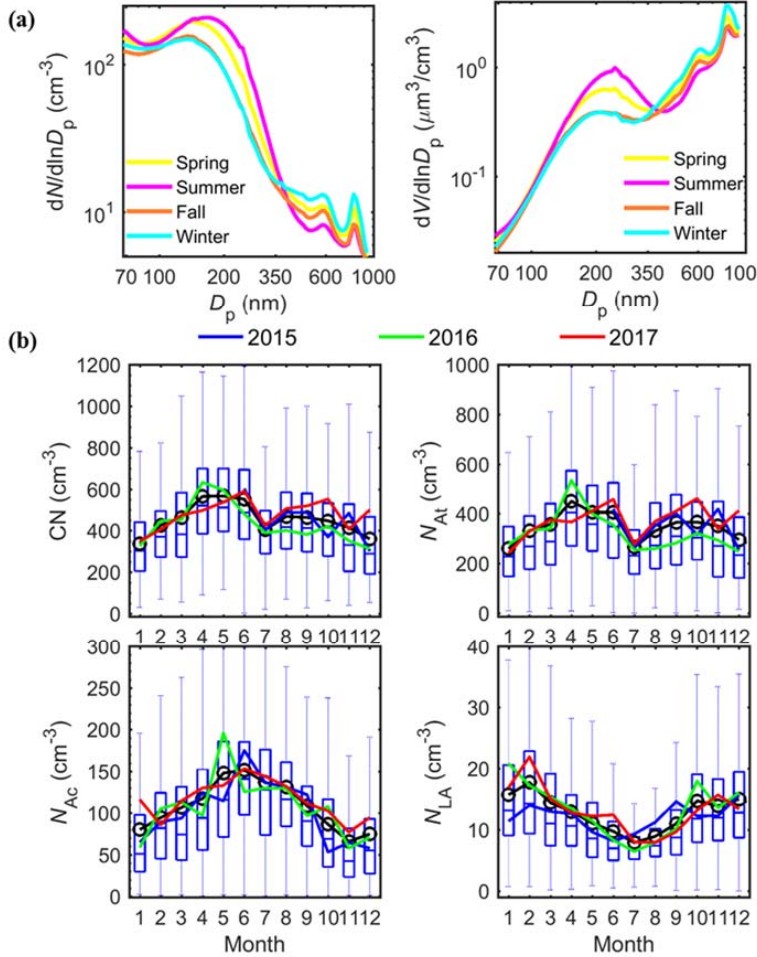

**Figure 4: Annual and seasonal variations of aerosol size distributions at the ENA site from 2015 to 2017.** (a) Seasonal-averaged number and volume distribution; (b) similar to Fig 2a, but for total aerosol number $CN$, and the number concentrations of At, Ac, and LA modes ($N_{At}$, $N_{Ac}$ and $N_{LA}$).



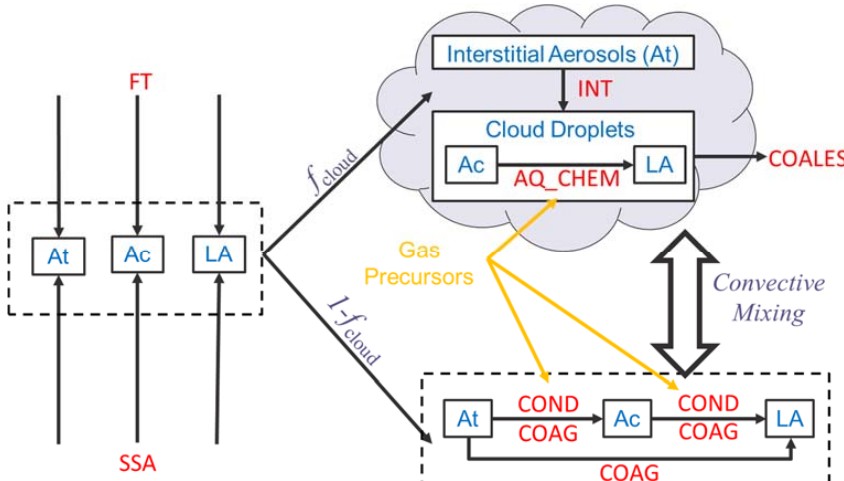

**Figure 5. Potential key controlling processes of MBL aerosol number concentrations considered in this study.**



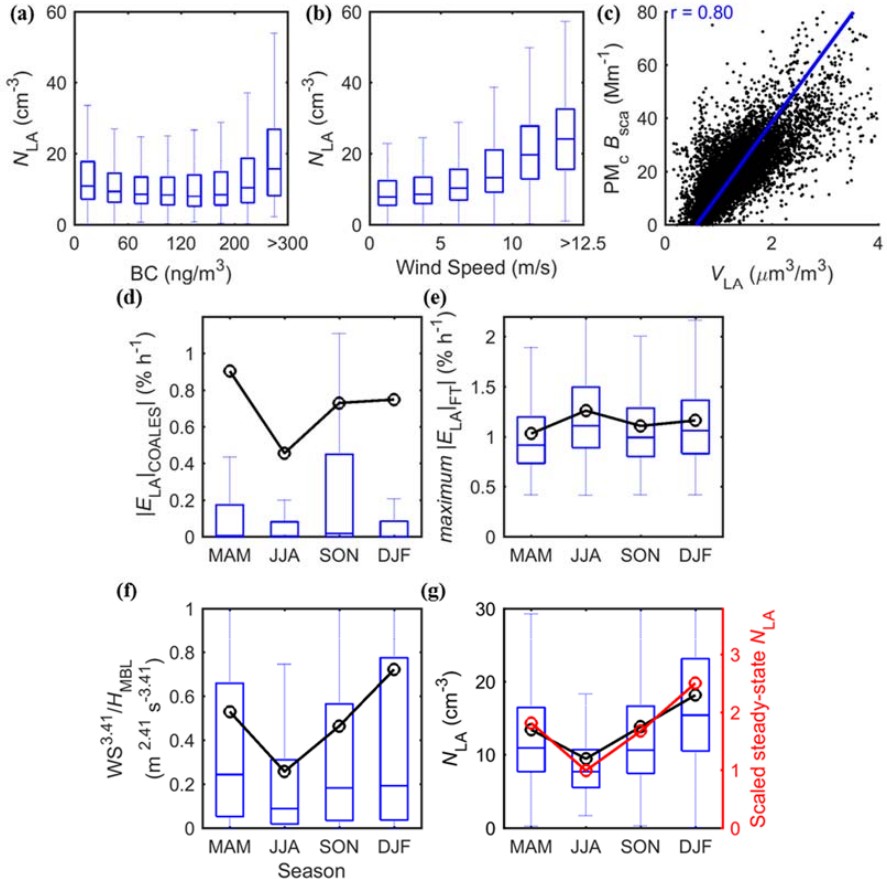

**Figure 6. Evidences of key controlling processes of LA mode as SSA and coalescence scavenging.** (a,b) Dependencies of $N_{LA}$ with BC and WS for data in 2015 to 2017; (c) correlation between $V_{LA}$ and $PM_c$ $B_{sca}$ for data in 2015 to 2017. The value of r given referred to the Pearson correlation coefficient, while the regression line based on York et al. (2004) is also shown for reference. (d,e) Estimated $N_{LA}$ sinking efficiency due to (d) coalescence scavenging and (e) dilution of FT entrainment. (f) Indicators of the major $N_{LA}$ source of SSA, and (g) the corresponding scaled ratios in comparison with observed $N_{LA}$ seasonal patterns. Data shown in (d-g) are from Sept. 2016 to Aug. 2017. The whispers and boxes indicated the 90th, 75th, median, 25th and 10th percentile, respectively. The black circle and lines indicated overall means.





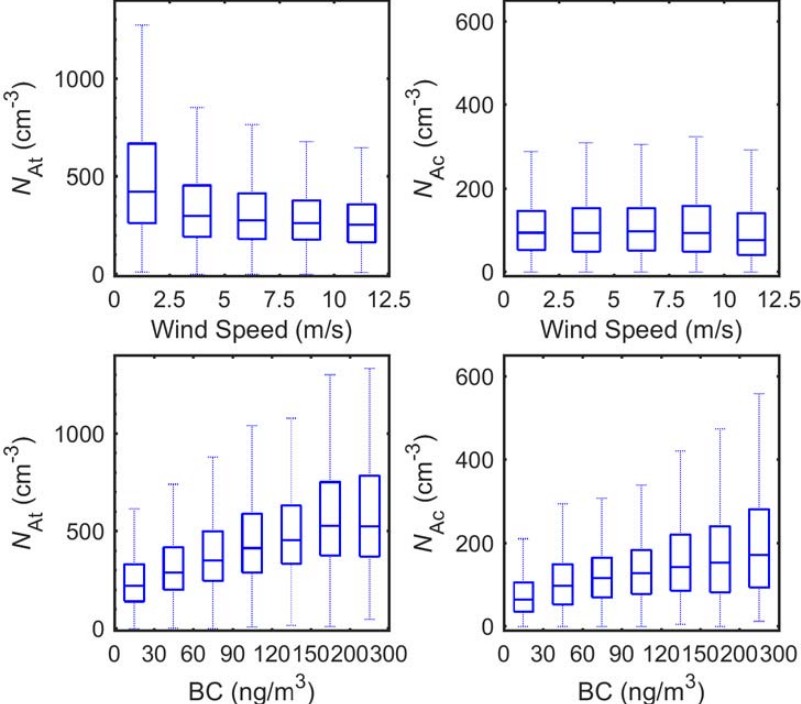

**Figure 7. Dependence of $N_{At}$ and $N_{Ac}$ on WS and BC in 2015 to 2017.** The whispers and boxes indicated the 90th, 75th, median, 25th and 10th percentile, respectively.





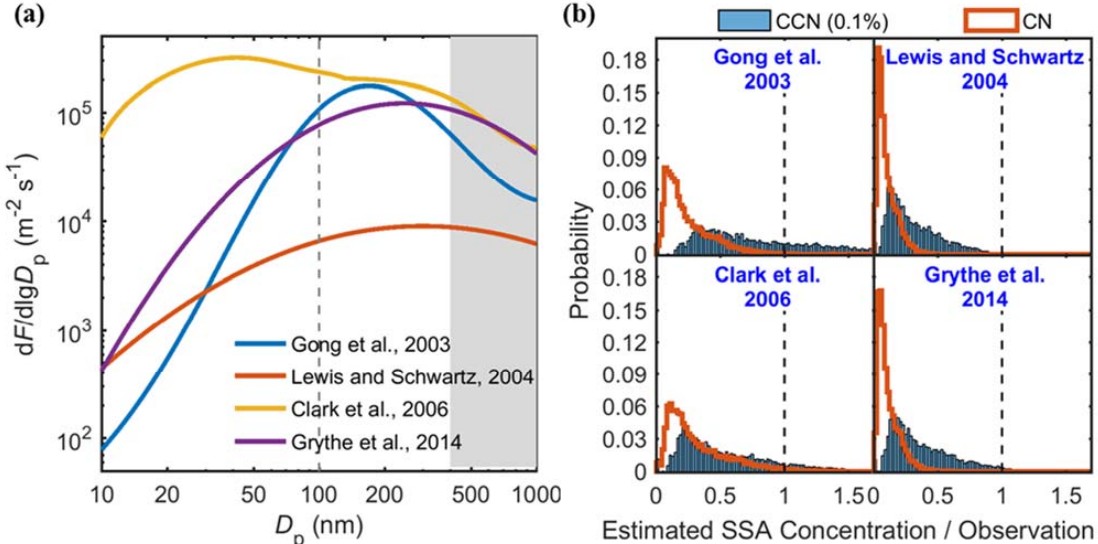

**Figure 8. Estimation of SSA contributions to *CN* and *CCN*(0.1 %).** (a) Previously published SSA production flux functions used here, and (b) SSA contribution to observed *CN* and *CCN*(0.1 %) estimated with each of the four SSA production flux functions.





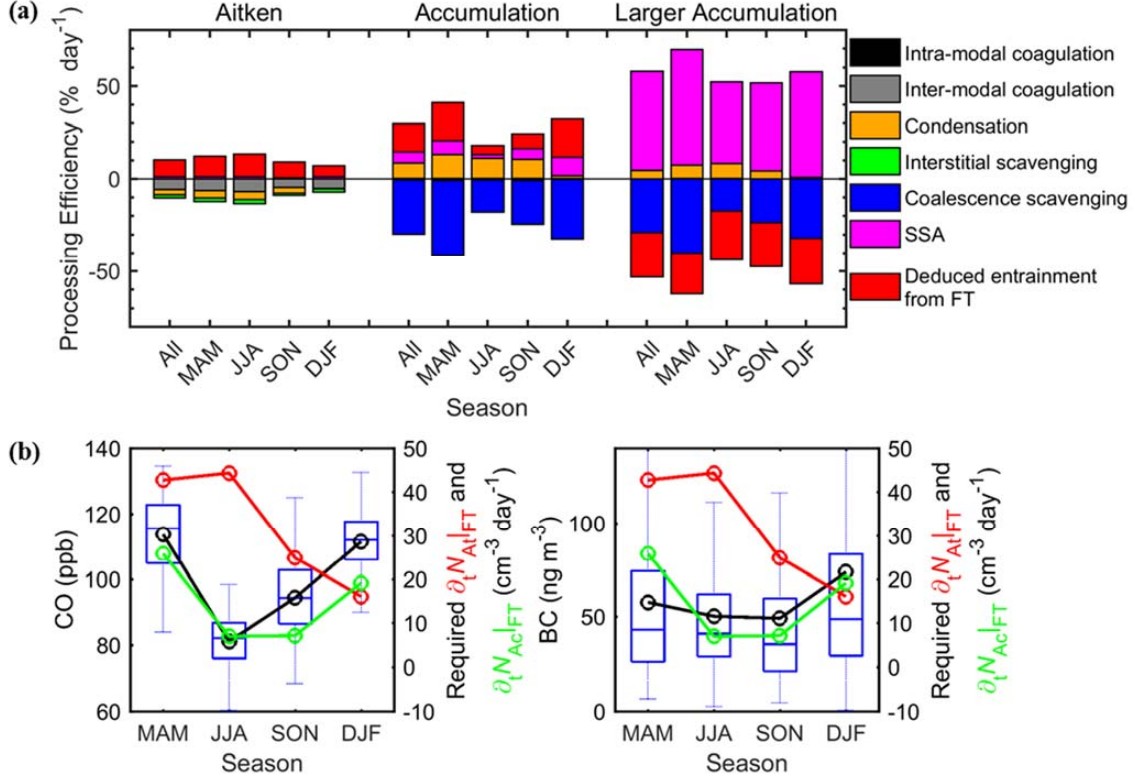

**Figure 9. Major controlling processes for each mode.** (a) Estimated secondary processing efficiency of each mode in different seasons. (b) Comparison of required seasonal-average FT entrainment rate to $N_{At}$ and $N_{Ac}$, with CO and BC. The whispers and boxes indicated the 90th, 75th, median, 25th and 10th percentile, respectively. The black circle and lines indicated overall means.



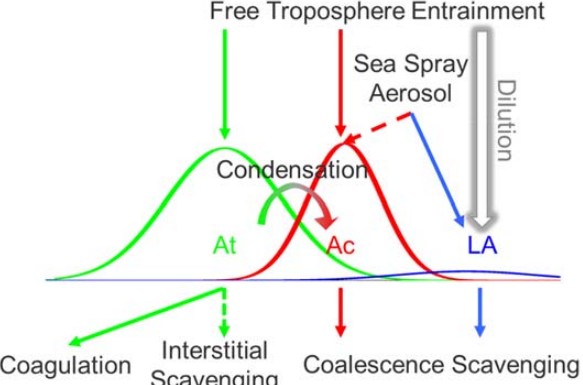

**Figure 10. Concept model of key controlling processes of MBL aerosol number concentrations for each mode at ENA.** Dash lines indicate the non-dominating but contributing processes. Negligible processes are not shown here.



**Table 1. Measurements of aerosol and cloud properties at the ENA site used in this study**

| Measurements | Symbol | Unit | Instruments | Time resolution | Measurement period |
|---|---|---|---|---|---|
| Total aerosol number concentration | $CN$ | cm$^{-3}$ | Condensation Particle Counter, Model 3772, TSI Incorporated, Shoreview, MN | 1 s | Oct. 2013 to Aug. 2014, June 2015 to present |
| Aerosol number size distribution from 60 nm to 1 μm | $dN/dlnD_p$ | cm$^{-3}$ | Ultra High Sensitivity Aerosol Spectrometer (UHSAS), DMT, Boulder, CO | 10 s | Feb. 2014 to present |
| $CCN$ number concentration at five super-saturations ($ss$)[a] | $CCN(ss)$ | cm$^{-3}$ | Cloud Condensation Nuclei counter, Model CCN-100, DMT, Boulder, CO | 1 s, ss level changes every ~12 min | Oct. 2013 to Apr. 2015, July 2016 to present |
| Aerosol absorbing coefficient | $B_{abs}$ | Mm$^{-1}$ | 3-wavelength Particle Soot Absorption Photometer (3λ-PSAP), Radiance Research, Seattle, WA, USA | 1 s for PSAP and 5 s for Nephelometer, inlet upper cut size changes between 1 μm and 10 μm every hour | Oct. 2013 to present |
| Aerosol scattering coefficient | $B_{sca}$ | Mm$^{-1}$ | Nephelometer, Model 3563, TSI Incorporated, Shoreview, MN | | Jan. 2014 to present |
| Trace gases of CO, NO$_2$ and H$_2$O | / | ppb | Gas Analyzer, Model 48C, Thermo Electron Corporation, Franklin, MA | 1 s | April 2015 to present |
| Trace gas of O$_3$ | / | ppb | Ozone monitor, Model 49i, Thermo Fisher Scientific Inc., Franklin, MA | 1 s | Oct. 2013 to present |
| Meteorological parameters [b] | / | / | ENA Aerosol Observing System (AOSMET, DOI: 10.5439/1025153) | 1 s | Jan. 2014 to present |
| MBL height [c] | $H_{MBL}$ | m | | | |
| Cloud thickness [c] | $h$ | M | Vertically pointing K-band cloud radar (KAZR); Ceilometer, Model CL31, Vaisala, Inc. (North America Support Office), Woburn, MA and Ceilometer CL31 | 16 s | Dec. 2014 to present |
| Cloudy time fraction [c] | $p_{cloud}$ | / | | | |
| Precipitation rate at cloud base [c] | $P_{CB}$ | mm h$^{-1}$ | | 30 min | Oct. 2015 to present |

[a] Measured at $ss$ levels of 0.1 %, 0.2 %, 0.5 % 0.8 % and 1 %.
[b] Including wind speed (WS) and wind direction (WD), temperature (T), pressure, relative humidity (RH), and rain rate at ground.
5    [c] See details in section 2.2.2.



**Table 2. Statistics of the fitted lognormal mode parameters of the number size distribution measured at the ENA site.** The numbers are shown as "mean (standard derivation)" for Sept. 2016 to Aug. 2017 and each of the four seasons during the one year period. Mode $D_p$ and mode σ are the mean and standard deviation of the fitted lognormal distribution of that mode, respectively.

| | | Annual | | Spring (MAM) | | Summer (JJA) | | Fall (SON) | | Winter (DJF) | |
|---|---|---|---|---|---|---|---|---|---|---|---|
| **Mode $N$ (cm⁻³)** | At | 330 | (239) | 386 | (250) | 360 | (226) | 301 | (265) | 273 | (190) |
| | Ac | 114 | (91) | 127 | (109) | 143 | (81) | 88 | (69) | 92 | (89) |
| | LA | 14 | (10) | 13 | (9) | 10 | (7) | 14 | (10) | 18 | (11) |
| **Mode $D_p$ (nm)** | Ac | 157 | (27) | 154 | (27) | 161 | (25) | 158 | (27) | 155 | (31) |
| | LA | 549 | (110) | 532 | (106) | 615 | (102) | 538 | (102) | 510 | (99) |
| **Mode σ** | Ac | 1.3 | (0.3) | 1.3 | (0.4) | 1.3 | (0.2) | 1.3 | (0.3) | 1.4 | (0.4) |
| | LA | 1.8 | (0.7) | 1.8 | (0.6) | 1.8 | (0.7) | 1.8 | (0.7) | 1.8 | (0.6) |
| **Occurrence (%)** | Ac | 85 | | 86 | | 93 | | 86 | | 73 | |
| | LA | 83 | | 79 | | 84 | | 84 | | 86 | |
| **Mode gap $D_p$ (nm)** | At ~ Ac | 101 | (35) | 100 | (34) | 93 | (27) | 104 | (32) | 109 | (44) |
| | Ac ~ LA | 490 | (104) | 480 | (91) | 545 | (111) | 470 | (96) | 452 | (90) |





**Table 3. Estimated terms of the governing equations for three modes using size distribution parameters in Table 2[a].**

| Progress description | | Process Rate Quantified | Process Rate (cm$^{-3}$ day$^{-1}$) | | | | |
|---|---|---|---|---|---|---|---|
| | | | Annual | Spring (MAM) | Summer (JJA) | Fall (SON) | Winter (DJF) |
| Intra-modal Coagulation | At+At→Ac | $\partial_t N_{Ac}|_{COAG}$ | 0.3 | 0.4 | 0.4 | 0.2 | 0.2 |
| | Ac+Ac→LA | $\partial_t N_{LA}|_{COAG}$ | 0.02 | 0.02 | 0.02 | 0.01 | 0.01 |
| Inter-modal Coagulation | At+Ac→Ac | $-\partial_t N_{At}|_{COAG}$ | 14.1 | 18.3 | 20.4 | 9.8 | 9.0 |
| | At+LA→LA | $-\partial_t N_{At}|_{COAG}$ | 4.5 | 5.2 | 3.7 | 4.1 | 4.6 |
| | Ac+LA→LA | $-\partial_t N_{Ac}|_{COAG}$ | 0.2 | 0.2 | 0.2 | 0.1 | 0.2 |
| Gas-phase Condensation from H$_2$SO$_4$[a] | At→Ac | $-\partial_t N_{At}|_{COND} = \partial_t N_{Ac}|_{COND}$ | 9.5 | 16.1 | 15.2 | 9.1 | 1.4 |
| | Ac→LA | $-\partial_t N_{Ac}|_{COND} = \partial_t N_{LA}|_{COND}$ | 0.6 | 0.7 | 0.6 | 0.5 | 0.6 |
| In-cloud Coagulation of Interstitial Aerosol[b] | At→Cloud Droplet (Ac and LA) | $-\partial_t N_{At}|_{INT}$ | 5.6 | 6.8 | 8.6 | 3.7 | 3.9 |
| Coalescence Scavenging | Cloud Droplet (Ac and LA)→Drizzling | $-\partial_t N_{Ac}|_{COALES}$ | 33.1 | 50.9 | 24.6 | 20.8 | 29.6 |
| | | $-\partial_t N_{LA}|_{COALES}$ | 4.0 | 5.4 | 1.6 | 3.3 | 5.8 |

[a] Gas-phase H$_2$SO$_4$ is assumed to be 1.35 ppt (Pandis et al., 1994); see sensitivity analysis in section 6.2.



**Table 4.** Parameters and results in estimation of SSA contribution to $N_{At}$ and $N_{Ac}$.

|  | **Annual** | | | **Spring** | | | **Summer** | | | **Fall** | | | **Winter** | | |
|---|---|---|---|---|---|---|---|---|---|---|---|---|---|---|---|
| $k_{INT,max}$[a] | 3.4 | | | 4.4 | | | 1.7 | | | 3.5 | | | 4.3 | | |
| $f_{SSA,\,Ac}$ (%) | 21 | ± | 18 | 19 | ± | 15 | 12 | ± | 13 | 24 | ± | 18 | 31 | ± | 22 |
| $f_{SSA,\,At}$[a] (%) | 10 | ± | 10 | 9 | ± | 8 | 8 | ± | 6 | 7 | ± | 9 | 16 | ± | 12 |

[a] Here only an upper limit of $k_{INT}$ without considering the condensation growth is estimated. Correspondingly, the $f_{SSA,\,At}$ is also expected to be an upper limit.

