# Peer review of "Marine boundary layer aerosol in Eastern North Atlantic: seasonal variations and key controlling processes"

_Atmospheric Chemistry and Physics, 2018_

## Referee Comment (RC1) · Anonymous Referee #1 · 26 Jul 2018

The paper provides a thorough analysis of the processes controlling the number concentration of the Aitken, accumulation, and sea spray aerosol modes in the eastern North Atlantic based on a several year data record from a site in the Azores. The conclusions that the free troposphere is a significant source of the Aitken and accumulation modes in the MBL and that sea spray aerosol makes up a small fraction of the total particle number at this site are significant and consistent with recently published papers. One intriguing result, if I am interpreting the analysis correctly, is that a significant impact of biogenic sulfur on the CCN population requires the flux of continental Aitken mode particles from the FT to the MBL.

Page 2, line 40: add the qualifier ". . ..long term observation IN THE ENA."

Page 4, line 3: change to ". . .the parameters of which ARE DERIVED from fitting"?

Page 4, line 35: The red trajectories in Fig.1 a, c, and d are all very similar, i.e., originating over the Arctic and passing over northern Canada. Why are they described as "air masses influenced by anthropogenic emissions from North America for fall and winter" and "contribution from Arctic" for Spring. Also – I don't see the "northern Europe air masses" in the trajectories for spring.

Figure 3.b2. and throughout: Figure 3.b2. clearly shows that what is termed here to be "Large Accumulation" mode is actually the sea spray aerosol coarse mode. To be in line with what it actually is and with published literature, it would be more appropriate to call it the SSA, PMA (primary marine aerosol), or primary aerosol mode.

Page 6, lines 23 – 24: The Ac mode Dp is 161 +/- 25 in summer and 155 +/- 31 in winter. Does the Ac mode really have a larger Dp in summer than winter given the fairly large standard deviations of the mean Dp?

Table 1: There is no instrument listed for MBL height or precipitation rate – unless they are included in the "Vertically pointing K-band. . .." list of instruments.

Table 2: Why aren't modal volumes included in the table – especially since they are referred to in the text (e.g., page 6, line 25).

Page 11, Line 9: should be Figure 6c.

Page 11, Lines 10 – 12: Has a volume mode with a diameter of 0.6 to 0.8 um ever been observed in the remote marine boundary layer? It is not clear why it is discussed here as a possibility and why the "LA" mode is not simply called the "SSA" mode.

Page 11, Lines 15 – 16: There are many, many published papers that establish that MBL supermicron particles are dominated by SSA. Why is it being debated/emphasized here?

Page 12, Lines 7 – 8: There is no need to invoke a lack of correlation of Nat or Nac with wind speed to conclude that SSA is a minor contribution to those two modes. Figure 3 is evidence enough.

Figure 8: Should make it clear in the caption that "(0.1%)" refers to supersaturation level.

Page 12, Line 33: should be ". . .fraction is consistent", not "in consistent".

Page 13, Line 2: "The SSA number concentration. . ." What number concentration is being referred to here? The present paper or Quinn et al., 2017? Page 13, Lines 3 – 7: Please clarify what the "above estimation" is. Numbering the equations and referring to them by number would help. Also, please define the f_ac,SSA and f_at,SSA terms. Are these the flux of SSA in the accumulation and Aitken modes, respectively?

Page 14, Lines 13 - 14: Please provide previously published fluxes of DMS in the ENA compared to the remote Southern Ocean. Also – this sentence is incomplete.

Figure 9: What is meant by secondary processing rate? Isn't the SSA flux a primary process, i.e., direct mechanical production?

Figure 9b: This half of Figure 9 does not appear to be explained in the main text.

Figure S5: Please provide r^2 values for these correlations to support the conclusion given on page 15, lines 20 – 21.

Page 16, Lines 11 – 19: Is the correct interpretation here that Aitken mode particles measured at ENA are continentally derived, while the growth of those particles to CCN size in the MBL is due to biogenic H2SO4? This implies that for ocean ecosystems (at least in the ENA) to have a substantial influence on the MBL CCN population, there must be Aitken mode continental aerosol for the required condensation and growth to occur.

[Figure]

2018.

---

## Referee Comment (RC2) · Anonymous Referee #2 · 31 Aug 2018

This paper uses two years of observations from the ARM program and a process-based analysis to estimate the factors that are most significant in defining the marine aerosol size distribution at Graciosa Island in the Azores. I find the approach to be systematic and reasonable, the large data set to be valuable and the results to be useful. I have a few little issues with some of the interpretation, but I think this work is worthy of publication in ACP subject to my following comments:

1) The paper reads better at the beginning. The grammar begins to suffer in various places later in the paper. The authors need to carefully read and correct the grammar where needed.

2) In the abstract and final paragraph (Page 16, lines 11-19), the authors say that the free troposphere (FT) is the major source of CCN to the marine boundary layer (MBL) via direct insertion and insertion of Aitken particles that grow via condensation in the MBL to CCN size; the latter being responsible for about 60% of the CCN in summer. That is a nice result. The authors then say that DMS oxidation is responsible. We cannot deny the likelihood of a contribution from DMS, but we know from work over the past few years that organics may play a significant role in Aitken particle growth. For example, in the Arctic we find growth of Aitken particles in the summer is related to (non-DMS) marine-derived organics at least as much and perhaps more than marine-derived sulphate (e.g. papers in the NETCARE special issue of ACP; work done at the Zeppelin Observatory; Willis et al., GRL, 2016; Burkart et al., GRL, 2017; Mungell et al., PNAS, 2017). The Arctic MBL differs from the Pacific and Atlantic MBLs in some respects, but there are similarities and various groups (O'Dowd et al.; Prather et al., etc.) have pointed to (non-DMS related) organics in Aitken particles in the MBL. The authors need to present a slightly more objective summary on this issue. I'm not suggesting to rule out DMS, but don't rule out the possibility of other organic components.

3) There is no mention of emissions from ships? What reasons do you have for excluding the possibility of shipping emissions? Related to ships and to the aqueous-phase chemistry and NPF processes, are SO2 concentrations measured at the ARM site? Have you any idea of what they might be? Are they too small to be routinely measured?

4) Abstract, lines 20-21 – Define the size ranges for At, Ac and LA modes here.

5) Abstract, line 28 – How is "the estimate based on major sources and sinks" made?

6) Page 2, line 20 – For previous aerosol studies, consider Phinney et al., Deep Sea Research (2006) and Langley et al. ACP (2010).

7) Page 2, lines 23-24 – "cloud coalescence scavenging" or "coalescence scavenging in cloud" or something with cloud factored in.

8) Page 2, line 39 – Consider Leaitch et al., ACP (2010) that looks at two cases of aerosols and MBL clouds over the Western Atlantic. Also, they found the supersaturations to be between 0.1 and 0.2% depending on the aerosol concentrations.

9) Page 3, lines 26-27 – Indicate how the inlet has been evaluated for transmission losses of the larger particles (i.e. 5-10 um diameter).

10) Section 2.2.2 – Presumably, cloud base height and cloud thickness are used in the analysis, but at this point (Section 2.2.2) it is unclear why cloud properties are discussed. Maybe on page 3 (around lines 5-6) or at the beginning of 2.2.2?

11) Page 5, line 23 – due to changes in tropopause height.

12) Your BC is measured from estimated absorption based on a filter transmission technique. The standard reference to this is Equivalent Back Carbon or EBC (Petzold et al., 2013) rather than BC. You should adopt that nomenclature.

13) Page 6, line 2 - define "Pcb"

14) Page 6, line 35 - Particles "activate" and droplets "nucleate".

15) Page 7, line 3 – Dry deposition may be slower on average, but you are discussing processes that happen over a week to 10 days. Please discuss further.

16) Page 8, lines 32-34 - Using a limit of 100 nm diameter for the Ac particles, you eliminate the potential for some larger Aitken particles to grow into the Ac mode via S(IV) oxidation. Without relatively large amounts of SO2, it will be very difficult for the Ac particles to grow into the LA mode. While still a big ask, for relatively low SO2, as seems more likely in the MBL you have constrained, the probability of growing 80 nm particles to 120 nm is more likely. What precursor concentrations do you use for your analysis?

17) Page 11, line 8-9 – Is the strong correlation between N(LA) and WS a result of using equation 2?

18) Page 11, line 19 – "Given the large sizes of LA particles and that we have excluded dust, we do not. . ."

19) Page 11, line 21 – ". . .the concentration of LA particles from the FT is negligible. . ."

20) Page 12, line 7 – The decreasing At with increasing WS could indicate some wind-associated dilution of oceanic sources of At particles.

21) Page 14, lines 25026 - That does not mean there is no contribution from anthropogenic emissions. It could be a case of the contribution being high in spring and low in summer.

22) Page 15, line 19 – It is 70 nm in the figures, not 60nm. I assume 70 nm is correct due to the noise issues that appear to be common in the first one or two channels of the UHSAS.

23) Page 15, line 24 – Sources of LA particles are dominated by SSA.

24) Page 15, line 24 – "dilution by entrained FT air". In order to dilute the LA particles with FT air, presumably some of the LA particles must enter the FT. Could that be an important FT source somewhere downwind?

25) Page 16, lines 14-15 – Can marine emissions be a factor here also, if they are lofted above the MBL somewhere and return to the MBL somewhere else?
* * *

---

## Author Comment (AC1) · 9 Oct 2018

**Manuscript No.**: acp-2018-574

**Title**: Marine boundary layer aerosol in Eastern North Atlantic: seasonal variations and key controlling processes

5   We thank the anonymous referee #1 for his/her valuable and constructive comments/suggestions on our manuscript. We have revised the manuscript accordingly and please find our point-to-point responses below.

**Comments by Anonymous Referee #1:**

10   *General Comments:*

*The paper provides a thorough analysis of the processes controlling the number concentration of the Aitken, accumulation, and sea spray aerosol modes in the eastern North Atlantic based on a several year data record from a site in the Azores. The conclusions that the free troposphere is a significant source of the Aitken and accumulation modes in the MBL and that sea spray aerosol makes up a small fraction of the*

15   *total particle number at this site are significant and consistent with recently published papers. One intriguing result, if I am interpreting the analysis correctly, is that a significant impact of biogenic sulfur on the CCN population requires the flux of continental Aitken mode particles from the FT to the MBL.*

*Detailed Comments:*

20   *1. Page 2, line 40: add the qualifier ": : :.long term observation IN THE ENA."*

**Responses:** The expression has been corrected as suggested.

*2. Page 4, line 3: change to ": : :the parameters of which ARE DERIVED from fitting"?*

**Responses:** The expression has been corrected as suggested.

25

*3. Page 4, line 35: The red trajectories in Fig.1 a, c, and d are all very similar, i.e., originating over the Arctic and passing over northern Canada. Why are they described as "air masses influenced by anthropogenic emissions from North America for fall and winter" and "contribution from Arctic" for Spring. Also – I don't see the "northern Europe air masses" in the trajectories for spring.*

**Responses:**

The trajectories shown in Fig. 1 are average trajectories for each cluster. The individual trajectories are shown in Fig. R1. We've also added this figure as Fig. S3 in the updated SI, and referenced in the manuscript accordingly (see Page 5, Line 1-4).

[Figure]

**Fig. R1** *(added as Figure S3 in updated SI)* **Detailed trajectories for each cluster shown in Fig. 1.**

*4. Figure 3.b2. and throughout: Figure 3.b2. clearly shows that what is termed here to be "Large Accumulation" mode is actually the sea spray aerosol coarse mode. To be in line with what it actually is and with published literature, it would be more appropriate to call it the SSA, PMA (primary marine aerosol), or primary aerosol mode.*

5 **Responses:**

We agree, as shown in the Fig. 3b2, that the "Large Accumulation" mode is dominated by SSA and is essentially the sea spray aerosol coarse mode under vast majority of the cases. On the other hand, we don't want to simply call the mode "SSA" or "PMA" mode, without presenting any evidence to demonstrate the case (as shown in Fig. 6). We also note during some episodes (not shown in the manuscript), aged biomass

10 burning aerosol and dust likely also contributed substantially to the large accumulation mode. We have added one sentence following the figure as (see Page 6, Line 26-28):

"Based on the average volume size distributions (Fig. 3b2), the "large accumulation mode" is essentially the sea spray aerosol coarse mode under vast majorities of the conditions."

15 *5. Page 6, lines 23 – 24: The Ac mode Dp is 161 +/- 25 in summer and 155 +/- 31 in winter. Does the Ac mode really have a larger Dp in summer than winter given the fairly large standard deviations of the mean Dp?*

**Responses:**

The standard deviation represents the variation of mode diameter during each season. We note that the
20 mean, median, $25^{th}$ and $75^{th}$ percentiles of the Ac mode diameter all exhibit a higher value during the summer than that during winter (Fig. R2). The larger mode diameter is also evidenced by the seasonal average size distribution (Fig. 3a). We have modified the sentence to the following (see Page 6, Line 31-32):

"While there is substantial variation within each season, on average, the Ac mode exhibits higher number
25 concentration, larger mode $D_p$, and higher occurrence in summer than in winter (Table 2)."

[Figure]

**Fig. R2 Seasonal variation of Ac mode diameters.**

*6. Table 1: There is no instrument listed for MBL height or precipitation rate – unless they are included in the "Vertically pointing K-band: : :." list of instruments.*

Responses: They are indeed included in the "Vertically …" list. We've added some solid lines to Table 1 to make it clearer.

*7. Table 2: Why aren't modal volumes included in the table – especially since they are referred to in the text (e.g., page 6, line 25).*

Responses: The modal volume information are added to Table 2 as suggested. In accordance with the data, we also modified the statement into (see Page 6, Line 34):

"In contrast, LA mode shows opposite seasonal trends, with the number and volume concentrations in winter exceeded 1.5 times those in summer (Table 2)."

*8. Page 11, Line 9: should be Figure 6c.*

Responses: Fig. 6c was discussed later (Page 11, Line 14). Here we are indeed discussing about Fig. 6b.

*9. Page 11, Lines 10 – 12: Has a volume mode with a diameter of 0.6 to 0.8 um ever been observed in the remote marine boundary layer? It is not clear why it is discussed here as a possibility and why the "LA" mode is not simply called the "SSA" mode.*

Responses:

Given the prevalence of marine low clouds, the brief discussion here is simply to eliminate the possibility that large accumulation mode observed is a result of in-cloud production of sulfate and/or organics (Pandis et al., 1990; Meng and Seinfeld, 1994; Seinfeld and Pandis, 2016). We agree the "LA" is essentially sea spray aerosol coarse mode under vast majorities of the conditions. Please also see the response to comment #4.

*10. Page 11, Lines 15 – 16: There are many, many published papers that establish that MBL supermicron particles are dominated by SSA. Why is it being ebated/emphasized here?*

**Responses:**

Here we're not emphasizing that supermicron particles are dominated by SSA. On the contrary, we are using that as a premise. What we stated is that since LA mode ($D_p$ ~300 to 1000 nm) share the same source with supermicron aerosols ($D_p$ 1~10 μm), and since supermicron aerosols are dominated by SSA in remote MBL, thus we speculate that LA mode should also be dominated by SSA. We've rephrased the description to avoid such confusion as (see Page 11, Line 14-19):

"This is also supported by the strong correlation between $V_{LA}$ and PM$_c$ $B_{sca}$ (Fig. 6c). The PM$_c$ $B_{sca}$ is a surrogate for the supermicron mode (PM$_c$, $D_p$ 1~10 μm) volume concentration (section 2.2), while supermicron particles are dominated by SSA in remote MBL (Campuzano-Jost et al., 2003). Therefore, the strong correlation suggests that LA particles are also dominated by SSA."

*11. Page 12, Lines 7 – 8: There is no need to invoke a lack of correlation of Nat or Nac with wind speed to conclude that SSA is a minor contribution to those two modes. Figure 3 is evidence enough.*

**Responses:**

Fig. 3 does show the fitted LA mode has a minor contribution to At mode or AC mode size range. However, LA mode likely presents the coarse mode of the SSA, and many studies have suggested that source function of SSA extends down to Aitken mode size range. We think lack of correlation provide additional evidence for the minor contribution.

*12. Figure 8: Should make it clear in the caption that "(0.1%)" refers to supersaturation level.*

**Responses:** The expression has been corrected as suggested into (see Caption of Fig. 8):

**"Figure 8. Estimation of SSA contributions to *CN* and *CCN* (0.1%), namely CCN concentration at 0.1% supersaturation level."**

*13. Page 12, Line 33: should be ": : :fraction is consistent", not "in consistent".*

**Responses:** The expression has been corrected as suggested.

*14. Page 13, Line 2: "The SSA number concentration: : :" What number concentration is being referred to here? The present paper or Quinn et al., 2017?*

**Responses:**

Here we mean the derived number concentration following the approach in Quinn et al. (2017). The sentence has been changed into (see Page 13, Line 2-3):

"In that study, the size distribution of SSA was derived by fitting aerosol size distribution. If we follow the same approach (Quinn et al., 2017), the estimated SSA number concentration is actually $N_{LA}$ shown in this study, which represents 19 % of *CCN* (0.1 %)."

*15. Page 13, Lines 3 – 7: Please clarify what the "above estimation" is. Numbering the equations and referring to them by number would help. Also, please define the f_ac,SSA and f_at,SSA terms. Are these the flux of SSA in the accumulation and Aitken modes, respectively?*

**Responses:**

The sentence has been changed into (see Page 13, Line 5-9):

"Based on the estimated SSA contribution to *CN* and *CCN* (Eq. 8a and 8b), we can further estimate the SSA contribution to $N_{Ac}$ and $N_{At}$, $f_{Ac, SSA}$ and $f_{At, SSA}$, as:

$$f_{Ac,SSA} = (CCN(0.1\%)_{SSA} - N_{LA}) / N_{Ac}$$
$$f_{At,SSA} = (CN_{SSA} - CCN(0.1\%)_{SSA}) / N_{At}$$

and the corresponding annual mean $f_{Ac, SSA}$ and $f_{At, SSA}$ are 21 % and 10 %, respectively (Table 4)."

*16. Page 14, Lines 13 - 14: Please provide previously published fluxes of DMS in the ENA compared to the remote Southern Ocean. Also – this sentence is incomplete.*

**Responses:**

This sentence is based on Plate 2 in Kettle et al. (1999), so we can only give an estimated range. We've modified the expression into (Page 14, Line 9-11):

"Such difference is likely due to the much higher DMS sea surface concentration in ENA (~7.5 nM) than that in southern oceans (~2.5 nM) (Kettle et al., 1999), or due to the difference between observed and model-simulated aerosol size distributions, etc.."

*17. Figure 9: What is meant by secondary processing rate? Isn't the SSA flux a primary process, i.e., direct mechanical production?*

**Responses:** The word "secondary" is deleted throughout.

*18. Figure 9b: This half of Figure 9 does not appear to be explained in the main text.*

**Responses:** It was discussed in the second but last paragraph in section 6.2 (Page 14, Line 1-4), and section 6.3. We also added the citation in section 6.3 as (see Page 14, Line 20):

"$\partial_t N_{At}|_{FT}$ is higher in spring-summer while lower in fall-winter, and such seasonal variation is somewhat different from those of CO mixing ratio and EBC mass concentrations (Fig. 9b)."

*19. Figure S5: Please provide r^2 values for these correlations to support the conclusion given on page 15, lines 20 – 21.*

**Responses:** The values are added to Fig. S5 (see Fig. R3 below).

[Figure]

**Figure R3.** *(Fig. S6 in updated SI)* **Comparison of observed *CCN* concentrations with relevant modal number concentrations.** The black dash line is the 1:1 line shown for reference. The value of *r* given referred to the Pearson correlation coefficient, while the regression line based on York et al. (2004) is also shown for reference.

*20. Page 16, Lines 11 – 19: Is the correct interpretation here that Aitken mode particles measured at ENA are continentally derived, while the growth of those particles to CCN size in the MBL is due to biogenic H2SO4? This implies that for ocean ecosystems (at least in the ENA) to have a substantial influence on the MBL CCN population, there must be Aitken mode continental aerosol for the required condensation and growth to occur.*

**Responses:**

Both continental emission and NPF in the FT contribute to the Aitken mode population in FT. However, we cannot quantitively determine the contribution from each of the two sources. Nevertheless, Fig. 9b shows much higher entrained FT At mode particles in spring-summer than entrained CO and BC, suggesting an important contribution of NPF to FT Aitken mode at least during these two seasons. We've added this discussion (see Page 14, Line 37 to Page 15, Line 2):

"Contribution of SSA to the At mode is even smaller than it is to the Ac mode, and is estimated to be no larger than 10 % (Table 4). As a result, the entrainment of FT At mode particles represents the dominant source (Fig. 9a). $\partial_t N_{At}|_{FT}$ is higher in spring-summer while lower in fall-winter, and such seasonal variation is somewhat different from those of CO mixing ratio and EBC mass concentrations (Fig. 9b). These differences may be partially due to stronger new particles formation from biogenic precursors in the FT during spring and summer seasons (Sanchez et al., 2018). The strength of new particle formation is not correlated with CO or EBC concentrations, which are tracers for anthropogenic emissions. The contribution of NPF versus anthropogenic emissions to FT Aitken mode particles cannot be quantitatively determined using data presented here alone, and will be a subject of future study."

**References**

Campuzano-Jost, P., Clark, C. D., Maring, H., Covert, D. S., Howell, S., Kapustin, V., Clarke, K. A., Saltzman, E. S., and Hynes, A. J.: Near-Real-Time Measurement of Sea-Salt Aerosol during the SEAS Campaign: Comparison of Emission-Based Sodium Detection with an Aerosol Volatility Technique, Journal of Atmospheric and Oceanic Technology, 20, 1421-1430, 10.1175/1520-0426(2003)020<1421:nmosad>2.0.co;2, 2003.

Kettle, A. J., Andreae, M. O., Amouroux, D., Andreae, T. W., Bates, T. S., Berresheim, H., Bingemer, H., Boniforti, R., Curran, M. A. J., DiTullio, G. R., Helas, G., Jones, G. B., Keller, M. D., Kiene, R. P., Leck, C., Levasseur, M., Malin, G., Maspero, M., Matrai, P., McTaggart, A. R., Mihalopoulos, N., Nguyen, B. C., Novo, A., Putaud, J. P., Rapsomanikis, S., Roberts, G., Schebeske, G., Sharma, S., Simó, R., Staubes, R., Turner, S., and Uher, G.: A global database of sea surface dimethylsulfide

(DMS) measurements and a procedure to predict sea surface DMS as a function of latitude, longitude, and month, Global Biogeochemical Cycles, 13, 399-444, 10.1029/1999GB900004, 1999.

Meng, Z., and Seinfeld, J. H.: On the Source of the Submicrometer Droplet Mode of Urban and Regional Aerosols, Aerosol Science and Technology, 20, 253-265, 10.1080/02786829408959681, 1994.

5    Pandis, S. N., Seinfeld, J. H., and Pilinis, C.: Chemical composition differences in fog and cloud droplets of different sizes, Atmospheric Environment. Part A. General Topics, 24, 1957-1969, https://doi.org/10.1016/0960-1686(90)90529-V, 1990.

Quinn, P. K., Coffman, D. J., Johnson, J. E., Upchurch, L. M., and Bates, T. S.: Small fraction of marine cloud condensation nuclei made up of sea spray aerosol, Nature Geosci, advance online publication, 10.1038/ngeo3003 http://www.nature.com/ngeo/journal/vaop/ncurrent/abs/ngeo3003.html#supplementary-information, 2017.

10  Sanchez, K. J., Chen, C.-L., Russell, L. M., Betha, R., Liu, J., Price, D. J., Massoli, P., Ziemba, L. D., Crosbie, E. C., Moore, R. H., Müller, M., Schiller, S. A., Wisthaler, A., Lee, A. K. Y., Quinn, P. K., Bates, T. S., Porter, J., Bell, T. G., Saltzman, E. S., Vaillancourt, R. D., and Behrenfeld, M. J.: Substantial Seasonal Contribution of Observed Biogenic Sulfate Particles to Cloud Condensation Nuclei, Scientific Reports, 8, 3235, 10.1038/s41598-018-21590-9, 2018.

Seinfeld, J. H., and Pandis, S. N.: Atmospheric chemistry and physics: from air pollution to climate change, John Wiley & Sons,
15  2016.

---

## Author Comment (AC2) · 9 Oct 2018

**Manuscript No.**: acp-2018-574

**Title**: Marine boundary layer aerosol in Eastern North Atlantic: seasonal variations and key controlling processes

5 We thank the anonymous referee #2 for his/her valuable and constructive comments/suggestions on our manuscript. We have revised the manuscript accordingly and please find our point-to-point responses below.

**Comments by Anonymous Referee #2:**

10 *General Comments:*

*This paper uses two years of observations from the ARM program and a process-based analysis to estimate the factors that are most significant in defining the marine aerosol size distribution at Graciosa Island in the Azores. I find the approach to be systematic and reasonable, the large data set to be valuable and the results to be useful. I have a few little issues with some of the interpretation, but I think this work is worthy*
15 *of publication in ACP subject to my following comments:*

*Detailed Comments:*

*1. The paper reads better at the beginning. The grammar begins to suffer in various places later in the paper. The authors need to carefully read and correct the grammar where needed.*

20 **Responses:** We've read through the updated manuscript and corrected the grammar carefully.

*2. In the abstract and final paragraph (Page 16, lines 11-19), the authors say that the free troposphere (FT) is the major source of CCN to the marine boundary layer (MBL) via direct insertion and insertion of Aitken particles that grow via condensation in the MBL to CCN size; the latter being responsible for about 60% of*
25 *the CCN in summer. That is a nice result. The authors then say that DMS oxidation is responsible. We cannot deny the likelihood of a contribution from DMS, but we know from work over the past few years that organics may play a significant role in Aitken particle growth. For example, in the Arctic we find growth of Aitken particles in the summer is related to (non-DMS) marine-derived organics at least as much and perhaps more than marine derived sulphate (e.g. papers in the NETCARE special issue of ACP; work done*
30 *at the Zeppelin Observatory; Willis et al., GRL, 2016; Burkart et al., GRL, 2017; Mungell et al., PNAS, 2017). The Arctic MBL differs from the Pacific and Atlantic MBLs in some respects, but there are similarities and various groups (O'Dowd et al.; Prather et al., etc.) have pointed to (non-DMS related) organics in Aitken particles in the MBL. The authors need to present a slightly more objective summary on this issue. I'm not suggesting to rule out DMS, but don't rule out the possibility of other organic*
35 *components.*

**Responses:**

We thank the reviewer for raising this point. We agree that while the ENA MBL differs from the Arctic MBL in some respects, it is possible that organics play an important role in the growth of Aitken mode particles. We've added relevant discussions in the updated manuscript. For example, we've added a comment in section 4.2, the condensation part as (see Page 9, Line 32-33):

"Here we assume that $H_2SO_4$ is the dominant condensate. However, recent studies suggest that organics may play an important role in growth of particles inside MBL, and this is discussed later in section 6.2."

And we've added the discussion into section 6.2 as (see Page 14, Line 14-21):

"Common continental biogenic volatile organic compounds (BVOCs) such as isoprene and monoterpenes typically have very low mixing ratio, and SOA formation from these BVOCs is generally minor in remote marine environment (Kavouras and Stephanou, 2002; Arnold et al., 2009; Gantt et al., 2009; Myriokefalitakis et al., 2010). However, recent studies suggest photochemistry or heterogeneous oxidation at the sea surface microlayer may represent a substantial source of oxygenated gas-phase organic compounds (OVOCs), which potentially plays an important role in SOA formation and particle growth in the Arctic MBL (Burkart et al., 2017; Willis et al., 2017; Mungall et al., 2017). It is possible that the SOA formation from these OVOCs can contribute to the growth of Aitken mode particles in ENA as well. If so, the contribution to CCN by the growth of Aitken mode particles would be even higher than the estimate here, which is based on condensation of $H_2SO_4$ only."

*3. There is no mention of emissions from ships? What reasons do you have for excluding the possibility of shipping emissions? Related to ships and to the aqueous-phase chemistry and NPF processes, are SO2 concentrations measured at the ARM site? Have you any idea of what they might be? Are they too small to be routinely measured?*

**Responses:**

We thank the reviewer for this point. Data impacted by local ship emissions are screened out and not included in the analyses (SI S1). Langley et al. (2010) show that ship particle emissions, when present, can contribute substantially to particles and CCN. The contribution of the ship particle emissions averaged over large spatial area in remote marine boundary layer remains unclear, and therefore it is not directly treated in this study. As a result, the particles emitted by ships are implicitly grouped into the category of "entrained from the FT".

Several studies (Langley et al., 2010; Corbett and Fischbeck, 1997; Capaldo et al., 1999; Corbett et al., 2007; Wang et al., 2008; Johansson et al., 2017) have shown ship emissions represent a significant source of $SO_2$ in MBL. In this study, the concentrations of $SO_2$ and $H_2SO_4$ are estimated using DMS-$SO_2$-$H_2SO_4$ yields based an observation-based parameterization (Russell et al., 1994; Pandis et al., 1994). Therefore, $H_2SO_4$ formed from ship emitted $SO_2$, and its contribution to condensational particle growth is implicitly included.

We have clarified this in the revised manuscript. We've added the following in section S1 (Page 2, Line 11-15):

"With this filter, data impacted by local ship emissions are also screened out. Langley et al. (2010) shows that ship particle emissions, when present, can contribute substantially to particle and CCN concentration in

the MBL. That condition, if present in ENA, would also be screened out considering the high aerosol number concentration (1000 ~ 3500 cm$^{-3}$). The contribution of the ship particle emissions averaged over large spatial area in remote marine boundary layer remains unclear, therefore it is not directly treated in this study."

With regards to regional ship emissions, we've added relevant discussions in section 6.2 as (see Page 14, Line 22-26):

"Several studies (Langley et al., 2010; Corbett and Fischbeck, 1997; Capaldo et al., 1999; Corbett et al., 2007; Wang et al., 2008; Johansson et al., 2017) have shown that ship emissions represent a significant source of SO$_2$ in MBL. In this study, the concentrations of SO$_2$ and H$_2$SO$_4$ are estimated using DMS-SO$_2$-H$_2$SO$_4$ yields based an observation-based parameterization (Russell et al., 1994; Pandis et al., 1994). Therefore, H$_2$SO$_4$ formed from ship emitted SO$_2$, and its contribution to condensational particle growth is implicitly included."

Unfortunately, SO$_2$ measurement is not available at the ENA site. Here we assume NPF is rare inside MBL, and estimated SO$_2$ concentration is used to qualitatively demonstrate that the aqueous-phase chemistry has negligible influence on the particle number concentration of each mode (see responses to comment #16).

*4. Abstract, lines 20-21 – Define the size ranges for At, Ac and LA modes here.*

**Responses:** The information has been added as (see Page 1, Line 23-24):

"Submicron aerosol size distribution typically consists of three modes: Aitken (At, diameter $D_p$ < ~100 nm), Accumulation (Ac, $D_p$ within ~100 to ~300 nm), and Larger Accumulation (LA, $D_p$ > ~300 nm) modes,…".

*5. Abstract, line 28 – How is "the estimate based on major sources and sinks" made?*

**Responses:** We've changed the expression into "generally agrees with the steady-state concentration estimated from major sources and sink."

*6. Page 2, line 20 – For previous aerosol studies, consider Phinney et al., Deep Sea Research (2006) and Langley et al. ACP (2010).*

**Responses:** The references have been added as suggested.

*7. Page 2, lines 23-24 – "cloud coalescence scavenging" or "coalescence scavenging in cloud" or something with cloud factored in.*

**Responses:** The expression has been changed into "in-cloud coalescence scavenging" throughout as suggested.

*8. Page 2, line 39 – Consider Leaitch et al., ACP (2010) that looks at two cases of aerosols and MBL clouds over the Western Atlantic. Also, they found the supersaturations to be between 0.1 and 0.2% depending on the aerosol concentrations.*

**Responses:**

This paragraph focused on campaigns in Eastern North Atlantic only. We referred to the Leaitch et al., ACP (2010) paper in discussions of typical supersaturation levels for marine low clouds instead (Page 9, Line1-2):

"The maximum supersaturation near the cloud base where *CCN* activation occurs is typically 0.2 % for marine low clouds (Wood et al., 2012; Clarke and Kapustin, 2010; Leaitch et al., 2010)."

*9. Page 3, lines 26-27 – Indicate how the inlet has been evaluated for transmission losses of the larger particles (i.e. 5-10 um diameter).*

**Responses:**

The transmission losses for the 5-10 um size range are not corrected. However, as we only examine the relative trends of the coarse mode optical properties and their correlations with other aerosol properties, we do not expect this will affect the results or conclusions from this study.

We've clarified this by including following description in the revised manuscript (see Page 3, Line 35 to 37):

"Potential particle losses for large particles (i.e., in the diameter range of 5 ~10 μm) are not corrected. However, we do not expect the losses affect the relative trends of $PM_c$ $B_{sca}$ presented here (section 5), or the correlation among $PM_c$ $B_{sca}$ and $V_{LA}$ (Fig. 6c)."

*10. Section 2.2.2 – Presumably, cloud base height and cloud thickness are used in the analysis, but at this point (Section 2.2.2) it is unclear why cloud properties are discussed. Maybe on page 3 (around lines 5-6) or at the beginning of 2.2.2?*

**Responses:**

We've added a paragraph at the beginning of section 2.2.2 as (see Page 4, Line 3-4):

"The cloud and MBL properties are needed to estimate some of the key controlling processes that drives aerosol properties (see section 4 for more details). Key parameter needed included the MBL height, cloud thickness and cloud fraction."

*11. Page 5, line 23 – due to changes in tropopause height.*

**Responses:**

The expression has been modified into (see Page 5, Line 27-28):

"In contrast, CO and $O_3$ in ENA show a summer minimum and spring-winter maximum, which is consistent with the FT entrainment as the dominant source and corresponding seasonal variations in

tropopause height. This suggests minor contributions from local emissions and in-situ photochemistry (Parrish et al., 1998; Fischer et al., 2003; Mao and Talbot, 2004)."

*12. Your BC is measured from estimated absorption based on a filter transmission technique. The standard reference to this is Equivalent Back Carbon or EBC (Petzold et al., 2013) rather than BC. You should adopt that nomenclature.*

**Responses:**

The expression has been changed throughout as suggested, with a notion added in section 2 as (see Page 6, Lines 2-3):

"After these episodes are excluded, the equivalent black carbon (EBC, following the naming convection suggested by Petzold et al. (2013)) mass concentrations were estimated from $PM_1$ $B_{abs}$ with an assumed mass absorbing cross section of 7.5 $m^2\,g^{-1}$ at 529 nm (Bond et al., 2013)."

*13. Page 6, line 2 - define "Pcb"*

**Responses:** The definition (precipitation rate at cloud base) has been added as suggested.

*14. Page 6, line 35 - Particles "activate" and droplets "nucleate".*

**Responses:** The word "activated" has been deleted.

*15. Page 7, line 3 – Dry deposition may be slower on average, but you are discussing processes that happen over a week to 10 days. Please discuss further.*

**Responses:**

In previous work comparing the contribution of dry and wet depositions (Henzing et al., 2006; Wood et al., 2012; Mohrmann et al., 2018), the time discontinuity in wet depositions are already considered, as also did in our study by taking into account the cloud fraction. The results show that for submicron MBL aerosols, the contribution of dry deposition is very small, with a fractional contribution generally < 5% in total (dry and wet) removal processes (Henzing et al., 2006), and absolute loss rates "unlikely to exceed 2 $cm^{-3}\,d^{-1}$" (Wood et al., 2012) under most circumstances.

This point is further clarified in the manuscript as (see Page 7, Line 11-14):

"In addition, dry deposition is usually much slower compared to wet deposition for submicron particles, even after considering in the in-cloud possibility, $f_{cloud}$, which reflects the time and spatial discontinuity of the wet deposition processes (see the in-cloud scavenging part in section 4.2) (Lewis and Schwartz, 2004; Henzing et al., 2006; Wood et al., 2012; Mohrmann et al., 2018). Thus, it is neglected in further analysis."

*16. Page 8, lines 32-34 - Using a limit of 100 nm diameter for the Ac particles, you eliminate the potential for some larger Aitken particles to grow into the Ac mode via S(IV) oxidation. Without relatively large*

*amounts of SO2, it will be very difficult for the Ac particles to grow into the LA mode. While still a big ask, for relatively low SO2, as seems more likely in the MBL you have constrained, the probability of growing 80 nm particles to 120 nm is more likely. What precursor concentrations do you use for your analysis?*

**Responses:**

There may be some misunderstanding here. In this section we're discussing the aqueous-phase reactions inside cloud droplets. As has been clearly outlined in section 4 and depicted in Fig. 5, Aitken mode particles are not CCN and thus are not considered in aqueous-phase reactions. That is due to the much higher liquid water content and thus much higher aqueous-phase reaction rates in cloud droplets than in interstitial aerosols. In-cloud sulfate production can contribute to the growth of the Aitken mode aerosols, but that only occurs after the Aitken mode particles have already grown to *CCN* (i.e. into Ac mode size ranges) through condensation. We've added a paragraph at the start of this section to further clarify this (Page 9, Line 9-14):

"The aqueous-phase reaction (i.e., in-cloud production of sulfate) rate is positively related the liquid water content (Seinfeld and Pandis, 2016; Meng and Seinfeld, 1994; Pandis et al., 1990; Cheng et al., 2016). As the liquid water content of cloud droplets are orders-of-magnitude higher than that of interstitial aerosols, only aqueous-phase reactions inside the cloud droplets are considered here (Pandis et al., 1990). As a result, the aqueous-phase reactions only promote the growth of *CCN* (i.e., Ac and LA mode particles), the influence of aqueous phase reactions on the Aitken mode particles is neglected until they grow to *CCN* sizes through condensation (Hoppel et al., 1994; Pandis et al., 1990)."

The precursor concentrations are given in section 4.2, the condensation part. It was described as (see Page 9, Line 27-30):

(The annual mean $H_2SO_4$) "is assumed to be 1.0 ppt (Pandis et al., 1994), while being 1.4, 1.3, 1.1 and 0.2 ppt in spring, summer, fall and winter, respectively. This seasonal variation in $v_i$ is based on the monthly dimethyl sulfide (DMS) fluxes (assumed to be 7.0, 5.4, 2.9 and 1.0 $\mu$mol m$^{-2}$ day$^{-1}$ in spring, summer, fall and winter, respectively) given in previous studies in the North Atlantic Ocean (Tarrasón et al., 1995), and the proposed dependence of $H_2SO_4$ on DMS flux at the observed fluxes ranges (Pandis et al., 1994; Russell et al., 1994)."

*17. Page 11, line 8-9 – Is the strong correlation between N(LA) and WS a result of using equation 2?*

**Responses:** Not likely. The $N_{LA}$ is from fitting of the observation data, not calculated by Eq. 2.

*18. Page 11, line 19 – "Given the large sizes of LA particles and that we have excluded dust, we do not: : :"*

**Responses:** The expression has been corrected as suggested.

*19. Page 11, line 21 – ": : :the concentration of LA particles from the FT is negligible: : :"*

**Responses:** The expression has been corrected as suggested.

*20. Page 12, line 7 – The decreasing At with increasing WS could indicate some wind associated dilution of oceanic sources of At particles.*

**Responses:**

The MBL height likely play a more important role in the dilution of MBL aerosol than WS. A more plausible explanation may be that, the higher $N_{LA}$ at higher WS will increase the coagulation sink of Aitken mode aerosols. As coagulation is the dominant sink for Aitken mode aerosols, this can better explain the negative correlation of $N_{At}$ with WS.

We have added this potential explanation to the manuscript as (see Page 12, Line 7-9):

"Unlike $N_{LA}$, $N_{Ac}$ is independent of the WS, and $N_{At}$ decreases with increasing WS (Fig. 7), indicating relatively minor contributions from SSA to At and Ac modes. The negative correlation between $N_{At}$ and WS may be due to the enhanced $N_{LA}$ with increasing WS (Fig. 6), and thus enhanced coagulation loss for Aitken mode particles (see section 4.2 and section 6.3)."

*21. Page 14, lines 25-26 - That does not mean there is no contribution from anthropogenic emissions. It could be a case of the contribution being high in spring and low in summer.*

**Responses:**

We have further clarified this in the revised manuscript (see Page 14, Line 38 to Page 15, Line 2):

"$\partial_t N_{At}|_{FT}$ is higher in spring-summer while lower in fall-winter, and such seasonal variation is somewhat different from those of CO mixing ratio and EBC mass concentrations (Fig. 9b). These differences may be partially due to stronger new particles formation from biogenic precursors in the FT during spring and summer seasons (Sanchez et al., 2018). The strength of new particle formation is not correlated with CO or EBC concentrations, which are tracers for anthropogenic emissions. The contribution of NPF versus anthropogenic emissions in FT Aitken mode particles, cannot be quantitively determined using data presented here alone, and will be a subject of future study."

*22. Page 15, line 19 – It is 70 nm in the figures, not 60nm. I assume 70 nm is correct due to the noise issues that appear to be common in the first one or two channels of the UHSAS.*

**Responses:** The expression has been corrected to 70 nm throughout as suggested, and we added a notion below Table 1 as:

"[a] In fact the lower size limit of UHSAS is 60 nm. Here we used only data larger than 70 nm to avoid noises sometimes observed in the first several channels of the UHSAS."

*23. Page 15, line 24 – Sources of LA particles are dominated by SSA.*

**Responses:** The expression has been corrected as suggested.

*24. Page 15, line 24 – "dilution by entrained FT air". In order to dilute the LA particles with FT air, presumably some of the LA particles must enter the FT. Could that be an important FT source somewhere downwind?*

**Responses:**

We don't think so. The entrainment process is not an exchange between FT and MBL. Instead, FT air is entrained into and stays inside the MBL (Stull, 2012). The entrainment counteracts the large-scale substance in ENA, and help sustain the MBL height (Stull, 2012; Mohrmann et al., 2018; Wood and Bretherton, 2004). While we agree that some LA particles may enter the FT through deeper convection, it is worth noting that these convections are often associated with precipitation that efficiently removes LA mode particles. Therefore the influence of such process on FT particle population is likely negligible.

*25. Page 16, lines 14-15 – Can marine emissions be a factor here also, if they are lofted above the MBL somewhere and return to the MBL somewhere else?*

**Responses:** Please refer to our response to comment #24.